# DOMAIN-AWARE GRADIENT REUSE FOR ANOMALY DETECTION

## ABSTRACT

Anomaly detection relies on recognizing patterns that diverge from normal behavior, yet practical deployment is hampered by the inherent scarcity and heterogeneity of anomalous instances. These challenges prevent the training set from faithfully characterizing the underlying anomaly distribution, thereby fundamentally constraining the development of effective discriminative models for anomaly detection. Inspired by the observed consistency of gradient distributions across related domains during training, Domain-Aware Gradient Reuse (DAGR) is introduced as a transfer-learning framework that leverages this property. DAGR first learns an adaptive transformation by aligning source and target normal gradients, thereby neutralizing domain-specific effects. The same map then pushes forward the source anomalous gradients to computing estimated target anomalous gradients, which are combined with the true target normal gradients to guide the target-domain detector without labeled anomalies. This paper establishes a rigorous convergence proof that reinforces the framework's theoretical foundation. Comprehensive experiments on image and audio datasets demonstrate that the proposed method achieves state-of-the-art performance.

## 1 INTRODUCTION

Anomaly detection flags observations that deviate from the normal data manifold, underpinning applications such as automated fraud mitigation, early medical diagnosis, and industrial fault prediction. The surge in data volume and complexity therefore demands models with high representational capacity and robustness to distribution shifts, rendering deep neural networks the prevailing solution.

Although deep learning has advanced rapidly, anomaly detection is still impeded by two factors. First, the scarcity of anomalous samples leads to severe class imbalance. Second, the heterogeneity of anomalies ensures that any finite dataset represents only a small fraction of the anomaly space. Together, these limitations prevent the training data from accurately representing the underlying anomaly distribution and, in turn, hinder the convergence of discriminative models.

Prior work in anomaly detection spans supervised and unsupervised paradigms. Supervised methods address class imbalance and anomaly sparsity/heterogeneity via reweighting, augmentation, or generative synthesis; however, synthetic anomalies cover limited modes and promote overfitting, yielding poor open-set generalisation to previously unseen anomaly types. Unsupervised methods model the normal manifold and detect deviations, yet the absence of anomalous supervision hampers calibration and discriminability—especially for subtle anomalies or under distributional drift. This scarcity–diversity dilemma motivates exploring transfer learning when the dataset under study lacks anomaly labels, leveraging related datasets that provide labelled anomalies.

Partial Domain Adaptation (PDA) is a natural option in this setting: it aligns the source–target distributions of normal features, after which source anomaly labels supervise learning in the shared representation. From an optimisation perspective, mini-batch updates decompose into normal and anomalous gradient components. Supervised training on the source induces anomalous-gradient directions tailored to the source distribution; under domain shift, these directions need not benefit the target to the same extent. Moreover, in anomaly-detection deployments where the target provides only normal data (one-class condition), the anomalous component is missing, yielding an incomplete update signal. This motivates estimating the missing component in the target gradient space via a learned transport map from the source. Fig. 1 examines feasibility: across epochs, per-sample

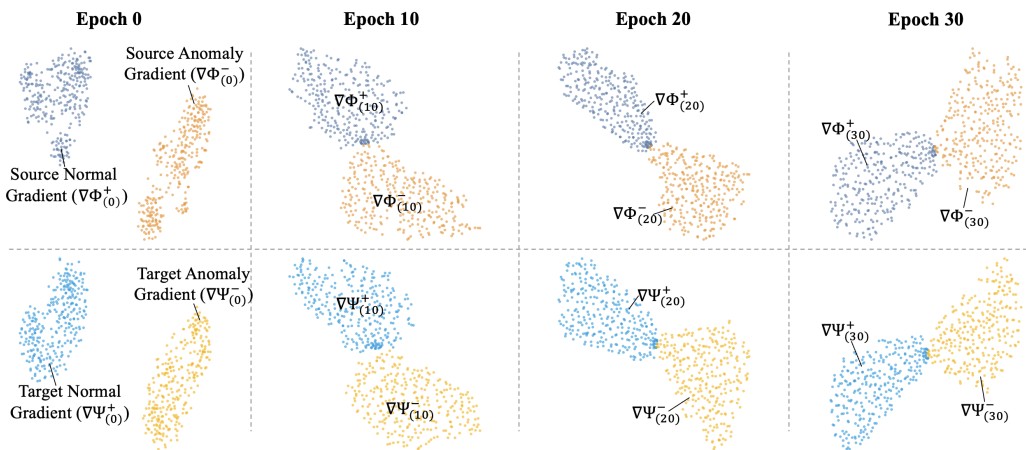

Figure 1: **Cross-Domain Gradient Distribution Consistency.** Each panel visualises, via t-SNE, the per-sample *gradient vectors* obtained at four training epochs (0, 10, 20, 30; left→right). **Top row:** source domain (Fan) showing normal ($\nabla\Phi^+$, blue) and anomalous ($\nabla\Phi^-$, orange) gradients. **Bottom row:** target domain (Pump) presenting normal gradients ($\nabla\Psi^+$, sky-blue) together with anomalous gradients estimated by the proposed mapping ($\nabla\hat{\Psi}^- = \mathcal{F}(\nabla\Phi^-)$, yellow). Across all epochs, the spatial arrangement of normal and anomalous manifolds in the two domains remains highly congruent, empirically supporting the assumption $\mathcal{P}(\nabla\Phi^+, \nabla\Phi^-) \approx \mathcal{P}(\nabla\Psi^+, \nabla\Psi^-)$ and thereby motivating the cross-domain gradient-reuse strategy.

gradients from two proximal domains form normal–anomalous manifolds with highly congruent (near-isometric) geometry, indicating that such a transport is learnable.

Building on this observation, Domain-Aware Gradient Reuse (DAGR) is introduced. DAGR first learns an adaptive transport map by aligning source and target normal gradients, thereby attenuating domain-specific components. The same map is then reused to project source anomalous gradients into the target space, producing estimated target anomalous gradients. In controlled evaluations where target anomalies are available for assessment, Fig. 2 shows that the mapped gradients closely overlap with the empirical target anomalous-gradient distribution across training epochs. By augmenting the target updates with this estimated anomalous component, DAGR guides the convergence of the target-domain detector without labelled anomalies.

This work proposes DAGR, a transfer-learning framework that remains effective even when the target domain contains no anomalous samples. Extensive experiments on image and audio benchmarks demonstrate state-of-the-art performance, while ablation studies isolate the contribution of each module. The appendix provides a complete convergence proof under stated assumptions, thereby giving the method a rigorous theoretical foundation.

## 2 RELATED WORK

Prior work on anomaly detection is grouped into three strands: augmentation-based supervised methods, unsupervised one-class modeling, and transfer learning.

### 2.1 SUPERVISED METHODS

Supervised anomaly detection commonly mitigates class imbalance via augmentation. Input-level transformations—geometric/photometric operations for images, time-warping and jittering for sequences, and graph-topology perturbations—expand the training distribution while preserving labels (Mumuni et al., 2024; Dang et al., 2023; Zhu et al., 2021). Generative augmentation employs GAN- or diffusion-based models to synthesise harder anomalous instances, enriching the minority class (Goodfellow et al., 2020; Cao et al., 2024). At the representation level, contrastive augmentation applies random masking or multi-view transformations with contrastive objectives to improve in-

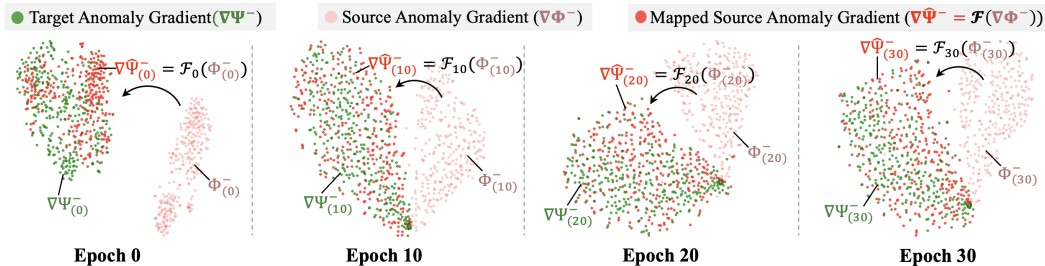

Figure 2: **Epoch-wise estimation of target anomalous gradients via cross-domain mapping.** Each panel shows a t-SNE embedding of *anomalous-sample* gradient vectors obtained at four training epochs (0, 10, 20, 30; panels arranged left→right). Green points represent the *true* target anomalous gradients $\nabla \Psi^-$; light-pink points depict the *source* anomalous gradients $\nabla \Phi^-$; red points are their epoch-specific images in the target space, $\nabla \hat{\Psi}^- = \mathcal{F}_{(e)}(\nabla \Phi^-)$. At each epoch, the mapped source-anomaly gradients nearly coincide with the true target-anomaly gradients, indicating that $\mathcal{F}_{(e)}$ provides an accurate *per-epoch* estimate of the target anomalous-gradient distribution.

variance and discrimination (Han et al., 2023; Zhou et al., 2022). Together, these strategies increase sample diversity without modifying ground-truth labels.

**Limitation.**    However, anomalous samples in training rarely reflect the true anomaly space. Even advanced augmentations generate limited variants and cannot bridge the semantic gap between observed and unseen anomalies, making generalisation beyond the augmentation manifold difficult.

## 2.2 Unsupervised Methods

Unsupervised anomaly detection learns normality from unlabelled data, using signals ranging from reconstruction fidelity to predictive objectives, representation discrimination, and density modelling. Reconstruction-based methods—Auto-Encoders, VAEs, GAN restorers, and diffusion decoders—identify anomalies by large residuals (Chen et al., 2018; An & Cho, 2015; Hussein et al., 2020; Wu et al., 2024). Self-supervised tasks such as future prediction, masked-signal recovery, and transformation discrimination extract intrinsic dynamics without labels (Venkatraman et al., 2015; Xie et al., 2023; Swarna et al., 2022). Contrastive learning compacts the normal manifold by attracting genuine instances and repelling perturbed views (Liang et al., 2022). Probabilistic density estimators—normalising flows and energy-based models—learn likelihoods so that low-density samples can be flagged (Garcia Satorras et al., 2021; Qin et al., 2022). These directions jointly approximate the normal manifold via reconstruction error, embedding compactness, and likelihood.

**Limitation.**    Because no anomalous instances participate in training, the learned boundary is inferred solely from normal data, often yielding overly broad decision regions and reduced precision on subtle or high-variance anomalies.

## 2.3 Transfer Learning for Anomaly Detection

Transfer learning is a viable strategy when the target dataset lacks anomalous samples, because a source domain enriched with labelled outliers can furnish the discriminative information that the target model requires. Transfer-based anomaly-detection research can be grouped into three lines of work. Partial Domain Adaptation (PDA) is the most widely adopted paradigm because it matches the practical setting where the target domain contains only normal data. PDA studies align cross-domain normal representations while suppressing source-only anomalies through class-importance weighting (Zhang et al., 2018), instance-level selection (Nguyen et al., 2023), or classifier-consistency (Jeong & Shin, 2020) strategies. Domain Adaptation (DA) assumes identical label spaces and exploits unlabeled target data to mitigate domain shift, typically using adversarial feature alignment (Chen et al., 2019), (Du et al., 2021), or self-supervised reconstruction constraints (Zhou et al., 2024). Domain Generalization (DG) trains without target data and pursues

domain-invariant features by applying meta-learning (Khoee et al., 2024) across multiple source domains, style or feature perturbations (Liu et al., 2024), or gradient-based regularization (Tang et al., 2021), thereby improving robustness to unseen environments.

**Limitation.** Although the alignment of normal representations mitigates interdomain shift, the optimisation signals derived from source-domain anomaly supervision remain domain-specific. Gradient directions that accelerate convergence in the source model do not necessarily align with the target optimisation landscape. Consequently, the transferred supervision provides limited discriminative guidance for the target detector and constrains further gains in anomaly-detection performance.

In summary, data augmentation improves class balance but does not address the unknown anomaly distribution. Unsupervised approaches avoid the need for anomaly labels, yet their precision remains limited because they lack anomalous guidance. Transfer-based methods also exhibit a critical shortcoming: the update directions induced by source anomalous supervision are optimised for the source distribution and are not guaranteed to benefit the target.

## 3 METHODS

This section details the proposed gradient-reuse framework. It first outlines the task setting and the motivation for exploiting source-domain information under severe anomaly scarcity. The subsequent subsection, "Cross-Domain Consistent Component Selection (CCCS)," explains how components that exhibit domain-invariant behavior are identified and preserved. Finally, "Adaptive Domain-Specific Perturbation Removal (ADPR)" describes how these components are leveraged to learn cross-domain gradients, enabling the estimation of target anomaly gradient. A rigorous convergence proof of DAGR is provided in the Appendix, establishing the theoretical soundness of the method.

### 3.1 MOTIVATION

Let the *source domain* $\mathcal{D}_s = \{(\mathbf{x}_i^s, y_i^s)\}_{i=1}^{N_s}$ contain both *normal* ($y = 0$) and *anomalous* ($y = 1$) instances, while the *target domain* $\mathcal{D}_t = \{\mathbf{x}_j^t\}_{j=1}^{N_t}$ is assumed to comprise normal data only. A $K$-layer deep network is considered, whose layerwise parameters are collected as

$$\Phi = \{\phi_1, \phi_2, \ldots, \phi_K\} \quad \text{and} \quad \Psi = \{\psi_1, \psi_2, \ldots, \psi_K\} \tag{1}$$

for the source and target models, respectively. At each training step we compute stochastic gradients $\nabla\Phi$ on the source mini-batch and $\nabla\Psi$ on the target mini-batch. Breaking the source gradient into class-conditioned components gives

$$\nabla\Phi = \nabla\Phi^+ + \nabla\Phi^-, \tag{2}$$

where $\nabla\Phi^+$ and $\nabla\Phi^-$ are from normal and anomalous samples, respectively. Because $\mathcal{D}_t$ lacks anomalies, only

$$\nabla\Psi^+ = \frac{1}{|\mathcal{B}_t|} \sum_{\mathbf{x}_j^t \in \mathcal{B}_t} \nabla_\Psi \mathcal{L}\big(f_\Psi(\mathbf{x}_j^t), 0\big) \tag{3}$$

is observable in the target domain, with $\mathcal{B}_t$ denoting the current target mini-batch.

**Empirical observation.** Figure 1 plots the distributions of $\nabla\Phi^+, \nabla\Phi^-, \nabla\Psi^+$ and $\nabla\Psi^-$ over training epochs. The divergence $\mathcal{D}\big(\nabla\Phi^+, \nabla\Phi^-\big) \approx \mathcal{D}\big(\nabla\Psi^+, \nabla\Psi^-\big)$ remains low, where $\mathcal{D}$ is instantiated as the 1-Wasserstein distance. This alignment suggests that both domains share a *domain-invariant* gradient component despite being collected in different environments.

**Gradient decomposition hypothesis.** We therefore posit that every mini-batch gradient can be decomposed into

$$\nabla\Phi = \Omega + \nabla\Phi_a, \quad \nabla\Psi = \Omega + \nabla\Psi_a, \tag{4}$$

where $\Omega$ encodes *cross-domain* knowledge that is useful for both domains; $\nabla\Phi_a$ and $\nabla\Psi_a$ capture *domain-specific perturbations*.

Because both domains provide abundant normal data, we learn a mapping

$$\mathcal{F} : \nabla\Phi^+ \longmapsto \Omega^+ \tag{5}$$

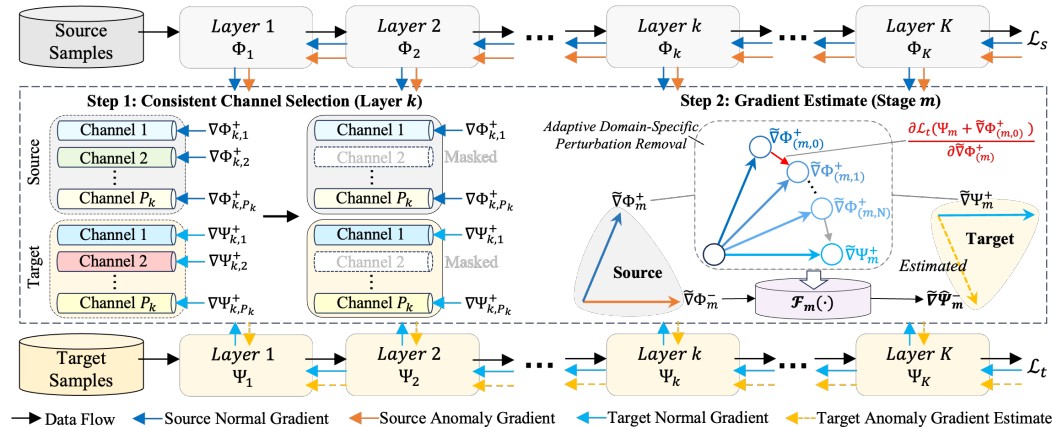

Figure 3: **Overall workflow of the proposed cross-domain gradient reuse framework.** The pipeline is executed on a *source network* ($\Phi_{1:K}$, top) and an architecturally identical *target network* ($\Psi_{1:K}$, bottom). **Step 1: Consistent Channel Selection.** For every layer $k$, the cosine similarity between the *normal-sample* gradients $\nabla\Phi^+_{k,p}$ and $\nabla\Psi^+_{k,p}$ of each channel $c_{k,p}$ is evaluated; channels whose similarity falls below the threshold $\gamma$ are masked (grey), leaving only domain-invariant components (solid colour). **Step 2: Adaptive Domain-Specific Perturbation Removal.** Given the masked source normal gradient $\tilde{\nabla}\Phi^+_m$ at outer stage $m$, an inner optimisation loop (blue trajectory) adapts it to the target loss, yielding the de-domainised estimate $\Omega^+_m = \mathcal{F}_m(\tilde{\nabla}\Phi^+_m)$. The same map $\mathcal{F}_m(\cdot)$ is then reused to transform the masked *anomalous* gradient, producing $\Omega^-_m$. Both components are aggregated as $\Omega_m = \Omega^+_m + \Omega^-_m$ and injected into the target network update (yellow dashed arrows), enabling anomaly knowledge transfer without exposing target data to anomalies.

that removes domain-specific noise from normal gradients. Under the *label-independent shift* assumption, Ben-David et al. (Ben-David et al., 2010) bound the target risk by the source risk plus the distribution divergence between domains. Coupled with the *Gradient Distribution Alignment* principle, this implies that the same $\mathcal{F}$ generalises to anomalous gradients:

$$\mathcal{F}(\nabla\Phi^-) \approx \Omega^-. \tag{6}$$

Aggregating $\Omega = \Omega^+ \cup \Omega^-$ yields a low-variance estimate of the domain-invariant descent direction.

**Transferring anomalous knowledge.** Finally, the target model is updated by

$$\psi_k \leftarrow \psi_k - \eta \cdot \Omega_k, \qquad k = 1, \ldots, K, \tag{7}$$

where $\eta$ is the learning rate. Equation equation 7 enables *implicit reuse* of *anomalous* gradients without exposing the target model to any anomalous data. We prove in Appendix that, under mild smoothness conditions (Gao et al., 2021a), the update rule in equation 7 reduces the target risk upper bound monotonically, thereby accelerating convergence.

The key insight is that *gradients—rather than features or logits—exhibit strong cross-domain regularities*. By denoising source gradients (equation 5), estimating the shared component $\Omega$ (equation 4), and injecting it into target updates (equation 7), our method transfers *anomalous knowledge* to a domain where no anomalies are observable. Formal justification is provided in Appendix A.1.

### 3.2 CROSS–DOMAIN CONSISTENT COMPONENT SELECTION

Although the source and target networks share an identical architecture, individual sub-modules (e.g., convolutional channels or Transformer heads) may specialise in *domain-specific* patterns. If such components participate in learning the de-domainisation map $\mathcal{F}$, the resulting estimate of the shared descent direction $\Omega$ would be biased. Hence, before training $\mathcal{F}$, we automatically identify and retain only those components whose behaviour is consistent across domains.

**Channel-wise gradient similarity.** Consider a $K$-layer CNN. Let $k \in \{1, \ldots, K\}$ index the layers and $\mathcal{C}_k = \{c_k^1, \ldots, c_k^{P_k}\}$ denote the $P_k$ output channels of layer $k$. For each channel $c_k^p \in \mathcal{C}_k$ we measure the *normal-sample* gradients in the source and target domains:

$$\nabla \Phi_{k,p}^+ = \frac{1}{|\mathcal{B}_s^+|} \sum_{\mathbf{x}_i^s \in \mathcal{B}_s^+} \nabla_{\phi_{k,p}} \mathcal{L}\big(f_\Phi(\mathbf{x}_i^s), 0\big), \tag{8}$$

$$\nabla \Psi_{k,p}^+ = \frac{1}{|\mathcal{B}_t|} \sum_{\mathbf{x}_j^t \in \mathcal{B}_t} \nabla_{\psi_{k,p}} \mathcal{L}\big(f_\Psi(\mathbf{x}_j^t), 0\big), \tag{9}$$

where $\phi_{k,p}$ (resp. $\psi_{k,p}$) collects the weights associated with channel $c_k^p$ in the source (resp. target) model. Their cosine similarity

$$\rho_{k,p} = \frac{\langle \nabla \Phi_{k,p}^+, \nabla \Psi_{k,p}^+ \rangle}{\|\nabla \Phi_{k,p}^+\|_2 \|\nabla \Psi_{k,p}^+\|_2} \in [-1, 1] \tag{10}$$

reflects the extent to which channel $c_k^p$ reacts similarly to *normal* data from both domains.

**Domain-invariant channel mask.** Given a global *channel–masking ratio* $\alpha \in (0, 1)$, the layer-wise threshold $\gamma_k$ is chosen as the $\alpha$-percentile of the cosine similarities $\{\rho_{k,p}\}_{p=1}^{C_k}$ in layer $k$:

$$\gamma_k = \text{Percentile}_\alpha\big(\{\rho_{k,p}\}_{p=1}^{C_k}\big). \tag{11}$$

**Binary mask.** Using the data-driven threshold equation 11, a binary mask is defined as

$$m_{k,p} = \mathbb{I}\big[\rho_{k,p} \geq \gamma_k\big], \qquad c_k^p \in \mathcal{C}_k, \tag{12}$$

where $\mathbb{I}[\cdot]$ is the indicator function. Channels with $\rho_{k,p} < \gamma_k$ are treated as domain-specific and *deactivated* by nullifying their gradients: $\tilde{\nabla}\Phi_{k,p}^+ = m_{k,p} \nabla\Phi_{k,p}^+$, $\tilde{\nabla}\Psi_{k,p}^+ = m_{k,p} \nabla\Psi_{k,p}^+$, $\tilde{\nabla}\Phi_{k,p}^- = m_{k,p} \nabla\Phi_{k,p}^-$. Aggregating over all layers yields the final masked gradients $\tilde{\nabla}\Phi^\pm$ and $\tilde{\nabla}\Psi^+$.

The filtered gradients are fed into Eq. equation 5:

$$\Omega^+ = \mathcal{F}(\tilde{\nabla}\Phi^+), \qquad \Omega^- \approx \mathcal{F}(\tilde{\nabla}\Phi^-). \tag{13}$$

By explicitly excising domain-specific channels, the variance of the shared estimate $\Omega$ is further reduced, which empirically accelerates convergence and stabilises the target update rule equation 7.

### 3.3 ADAPTIVE DOMAIN-SPECIFIC PERTURBATION REMOVAL

The masked normal gradients $\tilde{\nabla}\Phi^+ \in \mathbb{R}^D$ and $\tilde{\nabla}\Psi^+ \in \mathbb{R}^D$ are extremely high-dimensional ($D \approx 10^{6-9}$) and exhibit complex, *non-linear* cross-domain discrepancies. Simple statistics (e.g., means or linear projections) are therefore insufficient for extracting the shared component $\Omega^+$. We instead implement the adaptive cross-Domain gradient distiller $\mathcal{F}(\cdot)$ of Eq. equation 5 as a *gradient-based, end-to-end adaptive procedure* that removes domain-specific perturbations from source gradients by directly minimising the *target* loss.

**Outer–inner optimisation view.** At global training step $m \in \mathbb{N}$ let $\Phi_m$ and $\Psi_m$ denote the current source and target network parameters, and define the (normal-sample) source gradient $g_m^{(0)} = \tilde{\nabla}\Phi_m^+$. We treat $g_m^{(0)}$ as the *optimisable variable* and run an inner loop of $N$ steps to obtain its domain-invariant component $g_m^{(N)}$.

**Target-aligned inner loop.** Starting from the *fast weight*

$$\Psi_{m,0} = \Psi_m + \alpha g_m^{(0)}, \tag{14}$$

where $\alpha > 0$ is a small, fixed step size, we freeze the backbone parameters $\Psi$ and iteratively refine $g_m^{(n)}$ by descending the target loss $\mathcal{L}_t \triangleq \mathcal{L}(\mathcal{D}_t; \cdot)$:

$$g_m^{(n+1)} = g_m^{(n)} - \beta \nabla_g \mathcal{L}_t\big(\Psi_m + \alpha g_m^{(n)}\big), \tag{15}$$

where $n = 0, 1, \ldots, N-1$ and $\beta > 0$ denotes the inner-loop learning rate. The inner gradient $\nabla_g \mathcal{L}_t$ is computed over the current target mini-batch $\mathcal{B}_t$ and back-propagated *through* the fast weight construction in Eq. equation 14, thereby allowing $g_m^{(n)}$ to adapt to target-domain feedback.

After $N$ refinement steps we define

$$\Omega_m^+ = \mathcal{F}(\tilde{\nabla}\Phi_m^+) \triangleq g_m^{(N)}, \tag{16}$$

and inject $\Omega_m^+$ into the target update rule equation 7. Because the inner optimisation equation 15 is conditioned *solely* on target normal data, $\Omega_m^+$ is empirically free of domain-specific artefacts.

**Complexity.** A detailed time/space complexity discussion and implementation notes are provided in Appendix.

**Algorithmic summary.** The overall training alternates between *(i)* sampling a source normal mini-batch to obtain $\tilde{\nabla}\Phi_m^+$, *(ii)* executing the inner loop equation 14–equation 15 to produce $\Omega_m^+$, and *(iii)* updating the target parameters via Eq. equation 7.

**Gradient reuse.** Once the de-domainisation map $\mathcal{F}$ has been obtained via the inner loop in Eqs. equation 14–equation 16, it is *reused* to process the *anomalous* source gradients:

$$\Omega_m^- = \mathcal{F}(\tilde{\nabla}\Phi_m^-). \tag{17}$$

We then aggregate the normal and anomalous components,

$$\Omega_m = \Omega_m^+ + \Omega_m^-, \tag{18}$$

and apply the shared descent direction to the target network using the update rule of Eq. equation 7. In this way, *anomalous knowledge* is injected into the target model purely through gradient transfer, with no anomalous samples ever appearing in the target domain.

## 4 EXPERIMENTS

### 4.1 EXPERIMENTS SETUP

**Datasets and source–target protocol.** DAGR was evaluated on DCASE 2020 Task 2 (Koizumi et al., 2020) and DAGM (Wieler et al., 2007). Source–target pairs were formed between *adjacent domains* sharing sensing modality and generative mechanism—DCASE features motor-driven machines recorded under a common acoustic pipeline, while DAGM comprises homogeneous manufactured textures—ensuring a meaningful transfer basis. For DCASE, *Fan* is fixed as the source for stable in-domain performance and gradients; *Pump*, *ToyCar*, and *ToyConv.* serve as typical motor-driven targets with prominent motor signatures, whereas *Valve*, dominated by electromagnetic actuation and airflow pulses, and *Slider*, driven by reciprocating stage motion with weak motor harmonics, are included as *heterogeneity stress tests* to probe robustness under stronger cross-domain differences. For DAGM, *Class 2* is chosen as the source owing to its strongest in-domain performance; *Class 1*, *Class 3*, and *Class 6* are selected as more challenging targets, whereas *Class 4/5* are omitted because unsupervised baselines already saturate. In all settings, all labelled source data are available, each target exposes only $10\%$ of its normal samples with no anomalies, and a single source is applied per dataset without per-target tuning.

**Baseline.** To ensure a thorough comparison with current state-of-the-art approaches, four baseline categories were evaluated. The unsupervised group comprised General-AD (Sträter et al., 2024) and GLASS (Chen et al., 2024). The partial domain adaptation group consisted of PDA (Bai et al., 2024), CMKD (Zhou & Zhou, 2024), UniNet (Wei et al., 2025b), ANC (Zhang et al., 2024), JWO (Chen, 2024), PWAN (Wang et al., 2025) and MLWE (Wen et al., 2024). The domain generalisation group included BDC (Zhang et al., 2025b), DDDG (Zhang et al., 2025a), GGA (Ballas & Diou, 2025) and DKGPL (Wei et al., 2025a).

Table 1: AUROC (%) comparison on 8 benchmark datasets. Best per column in **bold**, second best is underlined.

| Methods | Source Domain | DCASE (Fan) | | | | | DAGM (Class 2) | | | Ave. |
|---------|---------------|------|--------|-------|-------|----------|---------|---------|---------|------|
| | Target Domain | Pump | Slider | Valve | ToyCar | ToyConv. | Class 1 | Class 3 | Class 6 | |
| Unsupervised | General-AD | 69.80 | 82.20 | 66.09 | 58.39 | 58.80 | 59.95 | 70.89 | 90.11 | 69.03 |
| Methods | GLASS | 65.93 | 88.37 | **67.42** | 63.22 | 59.76 | 90.98 | 80.22 | 70.04 | 73.45 |
| Partial | JWO | 41.32 | 64.99 | 64.40 | 65.01 | 54.73 | 55.71 | 64.53 | 81.45 | 61.52 |
| Domain | PWAN | 60.97 | 53.83 | 52.66 | 58.10 | 59.67 | 50.21 | 59.94 | 56.69 | 56.51 |
| Adaptation | MLWE | 42.75 | 63.30 | 49.59 | 54.46 | 63.35 | 49.31 | 47.63 | 50.62 | 52.63 |
| | CMKD | 53.47 | 66.68 | 56.69 | 48.49 | 54.93 | 54.93 | 78.55 | 94.51 | 63.53 |
| Domain | UniNet | 45.62 | 49.32 | 52.72 | 44.92 | 51.72 | 48.64 | 56.08 | 59.08 | 51.01 |
| Adaptation | ANC | 62.08 | 45.19 | 51.55 | 68.10 | 76.86 | 56.89 | 57.63 | 51.96 | 58.78 |
| | FFTAT | 57.72 | 71.06 | 55.92 | 64.42 | 57.15 | 58.38 | 80.59 | 86.04 | 66.41 |
| Domain | GGA | 69.26 | 76.77 | 62.48 | 67.08 | 70.35 | 58.20 | 52.89 | 74.88 | 66.49 |
| Generalization | BDC | 60.22 | 52.75 | 53.00 | 56.38 | 54.78 | 51.53 | 56.81 | 62.30 | 55.97 |
| | DDDG | 58.73 | 59.95 | 52.74 | 53.12 | 52.28 | 57.65 | 56.73 | 63.53 | 56.84 |
| | PMGDG | 68.27 | 45.55 | 54.87 | 56.45 | 60.21 | 51.53 | 44.72 | 71.44 | 56.63 |
| | DKGPL | 50.62 | 63.87 | 52.57 | 64.62 | 60.79 | 50.15 | 60.63 | 58.72 | 57.75 |
| Proposed | DAGR | **83.42** | **88.96** | 55.62 | **72.08** | **79.14** | **93.31** | **81.69** | **95.27** | **81.19** |

**Evaluation Metrics.** Three metrics are considered to assess the performance of MDPE: AUROC, AUPRC, and Rec@$K$. **AUROC** (area under the ROC) quantifies the ability to distinguish positive from negative classes and is widely regarded as a stable, threshold-agnostic indicator of discrimination performance. **AUPRC** (area under the PRC) summarizes the trade-off between precision and recall across thresholds and is particularly informative under severe class imbalance. **Rec@$K$** (recall at rank $K$) measures the proportion of true anomalies retrieved among the top-$K$ ranked instances, where $K$ equals the number of anomalous samples in the test set. Owing to space constraints, AUROC is adopted as the primary metric and its results are reported in the main text.

The *source codes* and more results about *AUPRC*, *Rec@K* are given in the *supplementary materials*.

**Comparison with SOTA.** Table 1 reports AUROC on all targets. DAGR achieves the best score on seven of eight domains—every DCASE target except *Valve*—and ranks first on all three MVTec defects; *Valve* is non-stationary and cross-mechanism, hence outside our adjacent-domain scope. Averaged across benchmarks, it attains 81.19% AUROC, exceeding GLASS (73.45%) by +7.7 pp and GGA (66.49%) by +14.7 pp. The gains are consistent across acoustic targets (*Pump*, *Slider*, *ToyCar*, *ToyConveyor*) and visual targets (*Cable*, *Capsule*, *Hazelnut*), supporting the effectiveness of the proposed gradient-reuse strategy.

## 4.2 ABLATION STUDY

The ablation study investigates the contribution of each core component and the influence of channel-masking ratio ($\alpha$).

**Effectiveness of CCCS and ADPR** Table 2 compares the full DAGR model with four ablated variants. Dropping the Cross-Domain Consistent Component Selection (w/o CCCS) reduces mean AUROC by about 3 percentage points and mean AUPRC by about 6 points, showing that filtering out gradient-inconsistent channels offers a clear yet secondary gain. In contrast, eliminating the Adaptive Domain-Specific Perturbation Removal (w/o ADPR) causes a sharp decline of roughly 16 points in AUROC and 33 points in AUPRC, indicating that learning a cross-domain gradient transformation is essential for successful reuse of source information. Simply substituting ADPR with conventional feature alignment (w FA) or a linear gradient mapping (w LT) only partially restores performance; both alternatives still trail the complete model by more than 15 points in AUROC and more than 30 points in AUPRC on average. Beyond accuracy, these variants exhibit stable training dynamics: removing CCCS/ADPR reduces accuracy without inducing divergence, indicating that both modules act as variance-reducing regularisers for gradient-space updates. These results confirm that CCCS helps but ADPR is the primary driver of DAGR's effectiveness, and that sophisticated gradient-space adaptation is required to fully exploit source-domain knowledge.

Table 2: The results (AUROC, %) of the ablation study on different modules. The highest score is highlighted in **bold**.

| Model | Pump | | Slider | | Valve | | ToyCar | | ToyConveyor | |
|---|---|---|---|---|---|---|---|---|---|---|
| | AUROC | AUPRC | AUROC | AUPRC | AUROC | AUPRC | AUROC | AUPRC | AUROC | AUPRC |
| w/o CCCS | 82.65 | 51.86 | 87.20 | 86.79 | 48.95 | 15.93 | 70.46 | 69.72 | 74.14 | 57.58 |
| w/o ADPR | 57.79 | 28.14 | 69.86 | 39.92 | 54.91 | 16.08 | 55.03 | 25.52 | 62.07 | 36.23 |
| w FA | 61.53 | 25.61 | 70.86 | 42.23 | 42.39 | 22.98 | 66.52 | 34.66 | 59.45 | 33.80 |
| w LT | 59.97 | 20.98 | 55.93 | 31.05 | 42.07 | 24.08 | 60.65 | 29.19 | 64.35 | 34.89 |
| DAGR | **83.42** | **53.71** | **88.96** | **87.79** | **55.62** | **26.52** | **72.08** | **71.76** | **79.14** | **71.91** |

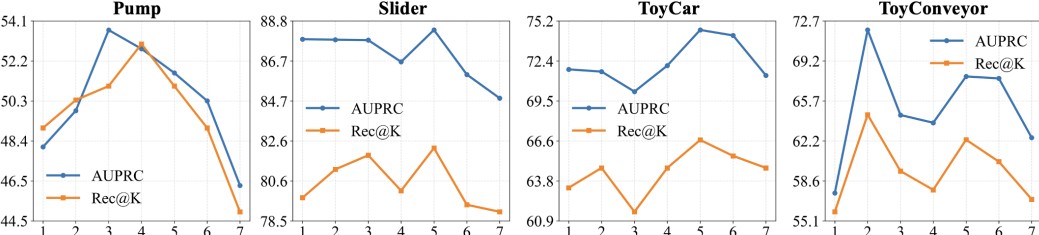

Figure 4: **Effect of channel-masking ratio ($\alpha$).** Detection performance (AUPRC and Rec@$K$) versus channel-masking ratio (%) on four representative benchmarks.

**Effect of the channel-masking ratio ($\alpha$).** Figure 4 reports AUPRC (green) and Recall@$K$ (red) as the masking ratio $\alpha$ is varied from 1% to 7% on four MIMII tasks. All curves rise when a small fraction of gradient-divergent channels is suppressed, peak in the 3% – 5% range, and decline thereafter. Peak values appear at 59.8% / 57.0% on Fan, 87.8% / 81.9% on Slider, and 74.6% / 66.7% on ToyCar. ToyConveyor reaches its first maximum at 2% and a secondary, gentler high near 5%. These results indicate that masking roughly 5% of channels achieves the best trade-off between noise removal and information retention; therefore $\alpha = 5\%$ is used in the remaining experiments.

### 4.3 DISCUSSION AND FUTURE WORK

DAGR delivers the strongest overall performance, reaching an average AUROC of 81.19% and ranking first on seven of eight targets, including all three DAGM classes, with consistent gains across acoustic and visual domains. The underperformance on *Valve* delineates the method's boundary rather than contradicting it. The *Fan* source exhibits quasi-stationary harmonic spectra from rotating parts, whereas *Valve* is dominated by non-stationary flow transients and a different physical process; the gradient-consistency premise is therefore not satisfied and transferability is limited. This clarifies the intended scope of DAGR: adjacent domains that share sensing modality and generative mechanism, such as motor-driven machinery and manufactured textures. For deployment, a proximity screen on normal-gradient geometry should be used ; when proximity falls below a threshold, the mapped anomalous component should be down-weighted or disabled, reverting to a conservative target-only update to avoid negative transfer. Accordingly, future work will enable proximity-aware gating by default and assess applicability under broader cross-domain shifts.

## 5 CONCLUSION

This paper presents Domain-Aware Gradient Reuse (DAGR), a transfer-learning framework that reinterprets domain adaptation as the selective reuse of source-domain gradients. By integrating gradient-consistency filtering with adaptive perturbation removal, DAGR provides both a formal convergence guarantee and a practical pathway to cross-domain generalisation. Extensive experiments on eight acoustic and visual anomaly detection benchmarks achieve state-of-the-art performance, showing that gradients, rather than features, can serve as an effective conduit for knowledge transfer. These results introduce a gradient-centric perspective to anomaly detection and open promising avenues for future adaptation strategies grounded in gradient compatibility.

## ETHICS STATEMENT

This work adheres to the ICLR Code of Ethics. Experiments rely on publicly available or appropriately licensed datasets; where data may contain personal or sensitive attributes, de-identification and license terms are respected, and no attempt is made to re-identify individuals. The study does not target or enable discriminatory or unsafe use; foreseeable dual-use risks are discussed and mitigation strategies (e.g., responsible release, robustness and bias checks) are described in the supplementary materials. No human-subject intervention, clinical decision-making, or deployment in safety-critical settings was conducted; any future deployment will follow applicable legal and institutional review requirements. Funding sources and potential conflicts of interest are disclosed. All procedures, data handling, and reporting were conducted with attention to privacy, fairness, and research integrity.

## REPRODUCIBILITY STATEMENT

Reproducibility has been prioritized. The main text specifies datasets and splits, preprocessing pipelines, model architectures, training schedules, and evaluation protocols, while exact hyperparameters, random seeds, and ablation configurations are provided in the appendix. An anonymized repository with source code, configuration files, and scripts to regenerate all tables and figures is supplied in the supplementary materials, including environment setup instructions and hardware requirements. For theoretical results, assumptions are stated in the paper and complete proofs are included in the appendix. External baselines are referenced with versions or commit hashes, and dataset licenses and checksums are reported to ensure fidelity. Together, these materials enable end-to-end reproduction of the reported results.

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

# A  APPENDIX

The supplementary material provides additional information, including the theoretical foundations of Domain-Aware Gradient Reuse (*Section A.1*), datasets descriptions (*Section A.2*), further comparative experiments (*Section A.3*) and disclosure of language model assistance (*Section A.4*). The source code for the proposed method is provided in the `code` directory.

## A.1  THEORETICAL FOUNDATIONS

In this section, **Supplementary A.1.1** demonstrates a *cross-class generalisation* property for the proposed de-domain mapping. After the mapping is trained *exclusively* on normal-class gradients, it is able to project source-domain abnormal gradients so that their distribution matches that of the (un-observed) target-domain abnormal gradients up to the same small tolerance level. **Supplementary A.1.2** leverages this result to study the optimisation trajectory of the Domain-aware Gradient Reuse (DAGR) algorithm. It is shown that, at every training step, the surrogate gradient used by DAGR differs from the exact target-domain gradient by a *uniform* and *time-independent* margin whose size is the sum of the tolerances proved in Supplementary A.1.1 and a bounded domain-specific perturbation term. Under standard smoothness and Polyak–Łojasiewicz conditions on the loss function, classical inexact-gradient descent theory (Bertsekas, 1999) then guarantees that the model parameters converge to a neighbourhood of the optimum whose radius is proportional to the square of this margin, and—crucially—this error does *not* accumulate over epochs.

Together, the two appendices provide a complete theoretical foundation for DAGR, simultaneously validating the reuse of source abnormal gradients and establishing the global convergence of the training procedure.

### A.1.1  PROOF OF CROSS-CLASS GENERALISATION

**Purpose.** This section proves that, under a *label-independent shift*, a mapping $\mathcal{F}$ learnt *solely* from **normal-class** gradients aligns **abnormal-class** gradients to the target domain with the same error upper-bound $\varepsilon$. This result substantiates the main-paper strategy of re-using source abnormal information—encoded in $\nabla\Phi^-$—even though the target domain contains no abnormal samples. This justifies the main-paper statement:

$$\mathcal{F}\big(\nabla\Phi^+\big)\approx\nabla\Psi^+ \quad\Longrightarrow\quad \mathcal{F}\big(\nabla\Phi^-\big)\approx\nabla\Psi^-,$$

and explains why the abnormal-class information contained in $\nabla\Phi^-$ can be safely reused in the target domain even when no abnormal samples are available there.

**Symbols and Decomposition.**    Let the $K$-layer source model have parameters $\Phi = \{\phi_1, \ldots, \phi_K\}$ and the target model $\Psi = \{\psi_1, \ldots, \psi_K\}$. For any mini-batch we obtain *expected* gradients $\nabla\Phi^+$, $\nabla\Phi^-$ (source, normal / abnormal) and $\nabla\Psi^+$ (target, normal). Following the main paper, *each* gradient splits into a domain-invariant component $\boldsymbol{\Omega}$ and a domain-specific perturbation:

$$\nabla\Phi \;=\; \boldsymbol{\Omega} + \nabla\Phi_a, \quad \nabla\Psi \;=\; \boldsymbol{\Omega} + \nabla\Psi_a.$$

We further write $\boldsymbol{\Omega}^+$ (normal) and $\boldsymbol{\Omega}^-$ (abnormal); note that $\boldsymbol{\Omega}^-$ exactly coincides with what the main text formerly denoted $\nabla\Phi_g^- = \nabla\Psi_g^-$, i.e. the abnormal but *domain-invariant* gradient component shared by both domains.

The random variables $G_\Phi^y$ and $G_\Psi^y$ ($y \in \{+, -\}$) represent per-sample gradients whose distributions are $\mathcal{P}_\Phi^y$ and $\mathcal{P}_\Psi^y$. A 1-Lipschitz distance $\mathcal{D}(\cdot, \cdot)$ —concretely the kernel Maximum Mean Discrepancy (MMD; see (Gretton et al., 2012))—measures distribution gaps.

**Label-Independent Shift Assumption.**    [Uniform Translation–Perturbation] There exists an *invertible* map $T : \mathbb{R}^d \to \mathbb{R}^d$ such that

$$G_\Psi^y \;=\; T\big(G_\Phi^y\big), \qquad \forall\, y \in \{+, -\}. \tag{A1}$$

Because the same $T$ applies to both labels, it transports the whole pair $(\boldsymbol{\Omega}^y, \nabla\Phi_a^y)$ to $(\boldsymbol{\Omega}^y, \nabla\Psi_a^y)$ without altering $\boldsymbol{\Omega}^y$. Empirical $t$-SNE plots in Fig. 1 verify this behaviour.

**Learning $F$ from Normal Gradients Only.**    With access to $\mathcal{P}_\Phi^+$ and $\mathcal{P}_\Psi^+$ we solve

$$\theta^\star = \arg\min_\theta\; \mathcal{D}\big(F_\theta(\mathcal{P}_\Phi^+), \mathcal{P}_\Psi^+\big), \tag{19}$$

producing a *de-domain* mapping $F_{\theta^\star}$. Its residual normal-class mismatch is

$$\varepsilon = \mathcal{D}\big(F_{\theta^\star}(\mathcal{P}_\Phi^+), \mathcal{P}_\Psi^+\big).$$

MMD ensures $\mathbb{E}[\varepsilon] = O(N^{-1/2})$ with $N$ normal samples (Gretton et al., 2012).

**Cross-Class Generalisation Theorem.**    Under Assumption A1 and with $F_{\theta^\star}$ from equation 19,

$$\mathcal{D}\big(F_{\theta^\star}(\mathcal{P}_\Phi^-), \mathcal{P}_\Psi^-\big) \;\le\; \varepsilon.$$

Triangle inequality yields

$$\mathcal{D}\big(F_{\theta^\star}(\mathcal{P}_\Phi^-), \mathcal{P}_\Psi^-\big) \le \mathcal{D}\big(F_{\theta^\star}(\mathcal{P}_\Phi^-), T(\mathcal{P}_\Phi^-)\big) + 0,$$

where the zero comes from Assumption A1. Because $\mathcal{D}$ is 1-Lipschitz, $\mathcal{D}(F_{\theta^\star}(z), T(z)) \le \|F_{\theta^\star}(z) - T(z)\|_2$. Let $\delta = \sup_z \|F_{\theta^\star}(z) - T(z)\|_2$; the same reasoning on the *normal* class gives $\varepsilon \le \delta$, hence the abnormal-class distance is bounded by $\varepsilon$.

**Relation to $\boldsymbol{\Omega}^-$.**    Because $T$ *preserves* the invariant part, $T(\boldsymbol{\Omega}^-) = \boldsymbol{\Omega}^-$. Applying $F_{\theta^\star}$ to source abnormal gradients gives

$$F_{\theta^\star}\big(\nabla\Phi^-\big) \;=\; F_{\theta^\star}\big(\boldsymbol{\Omega}^- + \nabla\Phi_a^-\big) \approx \boldsymbol{\Omega}^-,$$

up to error $\varepsilon$. Thus the mapped gradient contains (almost) exclusively the domain-invariant abnormal component $\boldsymbol{\Omega}^-$, meeting the requirement expressed in the main paper as $F(\nabla\Phi^-) = \nabla\Phi_g^- \approx \nabla\Psi_g^- = \boldsymbol{\Omega}^-$.

**Implications.**    By Theorem A.1.1,

$$\boldsymbol{\Omega}^- \;\approx\; F_{\theta^\star}\big(\nabla\Phi^-\big) =: \nabla\Phi_g^-,$$

which can be injected into the target update rule, despite the absence of abnormal target samples. Combining Ben-David's risk bound (Ben-David et al., 2006) with the fact that equation 19 shrinks the domain distance for *both* classes guarantees safe transfer. Gradient-space alignment has empirically achieved lower domain discrepancies than feature-space alignment (Gao et al., 2021b), supporting our choice of operating in gradient space.

### A.1.2   CONVERGENCE ANALYSIS OF DAGR

**Purpose.**   DAGR updates the **target-domain** parameters $\Psi = \{\psi_1, \ldots, \psi_K\}$ by a surrogate gradient $g_t = \mathcal{F}(\nabla\Phi^+) + F(\nabla\Phi^-)$, because *abnormal* target samples are absent. To justify its reliability, we prove that (i) $g_t$ deviates from the *true* target gradient $\nabla L(\Psi_t)$ by a *uniform* bound, and (ii) the resulting inexact-descent iterates $\{\Psi_t\}_{t\geq 0}$ converge to an $\mathcal{O}(\varepsilon)$ neighbourhood of the optimum without error accumulation.

**Notation (identical to Supplementary A.1.1)**

- Source parameters $\Phi = \{\phi_1, \ldots, \phi_K\}$, target parameters $\Psi$.
- Per-label gradients decompose as $\nabla\Phi^y = \mathbf{\Omega}^y + \nabla\Phi_a^y$ and $\nabla\Psi^y = \mathbf{\Omega}^y + \nabla\Psi_a^y$, $y \in \{+, -\}$.
- $F$ is the *de-domain* map from Supplementary A, trained with normal data; its precision satisfies $\|F(\nabla\Phi^y) - \mathbf{\Omega}^y\|_2 \leq \varepsilon = O(N^{-1/2})$.
- $L(\Psi)$ denotes the empirical target-domain loss, assumed $L$-smooth and $\mu$-PL.

**Assumptions   A1 (label-independent shift).**   There exists an invertible $T$ transporting all class-conditional gradient distributions (see Supplementary A).

**A2 (bounded perturbations).** $\|\nabla\Phi_a^y\|_2, \|\nabla\Psi_a^y\|_2 \leq B$ and $\mathbb{E}[\nabla\Phi_a^y] = \mathbb{E}[\nabla\Psi_a^y] = \mathbf{0}$.

**A3 (loss geometry).** $L(\cdot)$ is $L$-smooth and satisfies the PL-inequality with constant $\mu$.

**One-Step Gradient Error**   The true full gradient at step $t$ is $\nabla L(\Psi_t) = \mathbf{\Omega}^+ + \mathbf{\Omega}^- + \nabla\Psi_{a,t}^+ + \nabla\Psi_{a,t}^-$, whereas DAGR uses $g_t$. A direct triangle-inequality gives

$$\|g_t - \nabla L(\Psi_t)\|_2 \leq 2\varepsilon + 2B \ = \ \delta,$$

which is *constant in $t$*. Hence no per-iteration error growth can occur.

**Inexact-Gradient Descent Dynamics**   With step-size $\eta \leq 1/(2L)$, DAGR performs $\Psi_{t+1} = \Psi_t - \eta\,g_t$. Using standard inexact-descent analysis (Bertsekas, 1997) we obtain

$$L(\Psi_{t+1}) - L(\Psi^\star) \ \leq \ (1 - \eta\mu)\big[L(\Psi_t) - L(\Psi^\star)\big] \ + \ C\,\eta\,\delta^2,$$

where $C < 2$ is universal. Telescoping over $T$ steps and letting $T \to \infty$ yields the steady-state bound

$$\limsup_{t\to\infty} \big[L(\Psi_t) - L(\Psi^\star)\big] \ \leq \ \frac{C\,\delta^2}{\mu} \ = \ \mathcal{O}\big((\varepsilon + B)^2\big).$$

**Consequences**

- **Convergence.**  DAGR reaches an error ball whose radius contracts with $\varepsilon$; increasing normal-sample pairs tightens the bound.
- **No accumulation.**   $\delta$ is independent of $t$, so the error term in each step is constant—boundedness is preserved over indefinite epochs.
- **Practical implication.** When $B$ is empirically small (consistent dispersion of Fig. 1) and $N$ large, DAGR approaches the optimum as closely as exact gradient descent.

### A.2   DATASETS DESCRIPTIONS

**DCASE 2020 Task 2 Benchmark**   The DCASE 2020 Challenge Task 2 dataset(Koizumi et al., 2020) is a standard benchmark for unsupervised detection of anomalous sounds for machine condition monitoring. It features six distinct machine types: *ToyCar*, *ToyConveyor*, *Valve*, *Pump*, *Fan* and *Slide rail*. Each recording is a single-channel, 10-second audio clip sampled at 16 kHz. To leverage powerful feature extraction techniques from the vision domain, we first transform these one-dimensional audio signals into two-dimensional spectrograms, which reframes the original acoustic anomaly detection task into a visual anomaly detection problem. We designate the *Fan* subset as the source domain. The remaining five machine types—*ToyCar*, *ToyConveyor*, *Valve*, *Pump* and *Slide rail*—are treated as unseen target domains.

Table 3: AUPRC (%) comparison on 8 benchmark datasets. Best result per column is in **bold**, second best is underlined.

| Methods | Source Domain | DCASE (Fan) | | | | | DAGM (Class 2) | | | Ave. |
|---|---|---|---|---|---|---|---|---|---|---|
| | Target Domain | Pump | Slider | Valve | ToyCar | ToyConv. | Class 1 | Class 3 | Class 6 | |
| Unsupervised | General-AD | 66.40 | 78.79 | 62.77 | 48.90 | 40.02 | 32.81 | 37.64 | 83.72 | 58.63 |
| Methods | GLASS | **68.32** | 75.16 | **66.00** | 65.18 | 53.76 | 79.13 | 67.01 | 43.75 | **72.04** |
| Partial | JWO | 6.50 | 25.99 | 14.44 | 25.19 | 26.25 | 20.01 | 26.72 | 50.72 | 24.48 |
| Domain | PWAN | 16.10 | 20.36 | 14.68 | 17.32 | 24.04 | 16.07 | 20.07 | 22.32 | 18.87 |
| Adaptation | MLWE | 6.00 | 17.77 | 15.25 | 14.20 | 24.34 | 15.99 | 14.78 | 17.82 | 15.77 |
| | CMKD | 49.14 | 80.40 | 59.75 | 43.45 | 38.14 | 17.51 | 55.15 | 85.93 | 54.68 |
| Domain | UniNet | 12.00 | 14.84 | 20.17 | 11.61 | 19.31 | 16.86 | 18.54 | 23.01 | 17.04 |
| Adaptation | ANC | 12.44 | 12.09 | 15.92 | 21.63 | 35.37 | 16.49 | 22.12 | 16.67 | 19.09 |
| | FFAT | 16.53 | 37.38 | 26.26 | 32.19 | 34.99 | 22.98 | 41.42 | 70.45 | 35.27 |
| | GGA | 18.43 | 16.10 | 18.95 | 17.13 | 31.70 | 19.74 | 17.43 | 42.78 | 22.78 |
| | BDC | 12.15 | 15.58 | 15.69 | 19.43 | 18.02 | 16.66 | 21.08 | 24.19 | 17.85 |
| Domain | DDDG | 11.56 | 22.13 | 16.68 | 13.94 | 19.80 | 20.81 | 20.75 | 24.01 | 18.71 |
| Generalization | PMGDG | 11.94 | 12.34 | 14.99 | 14.24 | 21.75 | 16.75 | 13.91 | 35.94 | 17.73 |
| | DKGPL | 7.32 | 25.68 | 14.24 | 22.25 | 19.98 | 17.14 | 20.69 | 22.54 | 18.73 |
| Proposed | DAGR | 53.71 | **87.79** | 26.52 | **71.76** | **71.91** | **81.77** | **73.29** | **86.97** | 69.22 |

Table 4: Rec@K (%) comparison on 8 benchmark datasets. Best result per column is in **bold**, second best is underlined.

| Methods | Source Domain | DCASE (Fan) | | | | | DAGM (Class 2) | | | Ave. |
|---|---|---|---|---|---|---|---|---|---|---|
| | Target Domain | Pump | Slider | Valve | ToyCar | ToyConv. | Class 1 | Class 3 | Class 6 | |
| Unsupervised | General-AD | 58.41 | 72.93 | 63.52 | 49.00 | 41.29 | 32.00 | 41.33 | 73.33 | 56.46 |
| Methods | GLASS | **62.34** | 71.69 | **67.66** | 58.74 | 45.23 | 70.67 | 58.67 | 41.33 | **67.29** |
| Partial | JWO | 5.80 | 27.20 | 10.77 | 27.33 | 29.11 | 14.84 | 31.08 | 51.32 | 24.68 |
| Domain | PWAN | 23.19 | 26.40 | 20.77 | 18.00 | 27.22 | 15.79 | 23.68 | 23.68 | 22.34 |
| Adaptation | MLWE | 2.90 | 17.60 | 19.23 | 12.00 | 31.01 | 17.11 | 10.53 | 21.05 | 16.43 |
| | CMKD | 45.14 | 76.07 | 58.66 | 40.89 | 35.05 | 16.25 | 46.25 | 76.25 | 50.70 |
| Domain | UniNet | 10.77 | 16.00 | 23.47 | 10.00 | 22.78 | 19.74 | 19.74 | 23.68 | 18.27 |
| Adaptation | ANC | 17.65 | 6.40 | 17.69 | 23.33 | 35.44 | 11.84 | 26.32 | 15.79 | 19.31 |
| | FFTAT | 18.20 | 40.60 | 28.59 | 34.53 | 36.99 | 20.51 | 43.59 | 64.10 | 35.89 |
| | GGA | 20.29 | 18.40 | 21.54 | 22.00 | 32.91 | 21.05 | 17.57 | 44.74 | 24.81 |
| | BDC | 19.12 | 13.60 | 16.92 | 25.33 | 18.99 | 15.79 | 28.95 | 27.03 | 20.72 |
| Domain | DDDG | 12.70 | 28.80 | 17.69 | 18.00 | 24.05 | 18.92 | 18.92 | 24.86 | 20.49 |
| Generalization | PMGDG | 13.04 | 5.60 | 16.15 | 14.00 | 25.32 | 15.79 | 10.53 | 36.84 | 17.16 |
| | DKGPL | 4.41 | 28.00 | 15.38 | 25.33 | 19.62 | 15.79 | 18.42 | 23.68 | 18.83 |
| Proposed | DAGR | 53.02 | **81.16** | 26.96 | **66.29** | **64.46** | **72.69** | **67.75** | **82.27** | 64.33 |

**DAGM Dataset** The DAGM 2007 dataset(Wieler et al., 2007) is a widely-used benchmark designed for unsupervised visual anomaly detection. The dataset features 10 different classes of grayscale texture images, created to simulate various industrial surfaces. These images have a resolution of $512 \times 512$ pixels and contain several types of artificially generated defects. For each class, the dataset provides 1,000 defect-free images for training and 150 images for testing, which may or may not contain defects. In our experiment, all defects are treated as anomalies. The *Class 2* category served as the source domain, with the *Class 1*, *Class 3*, and *Class 6* categories acting as target domains.

## A.3 ADDITIONAL COMPARISON EXPERIMENTS

**Overall results.** Tables 3 and 4 report AUPRC and Rec@K on eight transfer tasks. The proposed **DAGR** attains the **best per-column score on 6/8 tasks** under both metrics, and ranks second in the overall average. Specifically, in AUPRC DAGR leads on *Slider, ToyCar, ToyConv., DAGM-Class 1/3/6*, and in Rec@K it again leads on the same six targets. The average, however, is depressed by a single outlier—the *Fan $\rightarrow$ Valve* transfer.

**Why the average is lower: the Valve outlier.** On *Valve*, DAGR is substantially weaker (AUPRC 26.52, Rec@K 26.96) than on the other targets. This behaviour is expected and consistent with the method's scope: DAGR explicitly relies on *domain proximity* diagnosed from *normal* data (Sec. §2/§3), i.e., the cross-domain normal-gradient geometry must be compatible enough to learn a reliable transport map. Most DCASE targets (*Pump, Slider, ToyCar, ToyConv.*) are *motor-driven rotating machinery*. Their acoustics are dominated by quasi-stationary tonal components at the rotation frequency and its sidebands, yielding stable spectra and slowly varying envelopes. *Valve* is qualitatively different: its sound is driven by *turbulence, flow transients and opening/closing events*, which are *non-stationary*, bursty and broadband. Consequently, the target normal-gradient subspace for Valve diverges from that of Fan, leading Stage-1 gating to mask many channels and Stage-2 to learn a weak transport; the mapped anomalous gradients are then down-weighted by the reliability scheme, effectively reverting to normal-only updates. In contrast, unsupervised baselines such as GLASS/General-AD train purely on target normals and are not affected by cross-domain incompatibility, hence their stronger scores on Valve.

**Quantifying the outlier effect.** The overall average in Tables 3 and 4 is a simple mean over eight targets. For **AUPRC**, DAGR's mean is 69.22%, but *removing the single Valve column for reference* raises it to **75.31**%, exceeding the best competing average (72.04%). For **Rec@K**, the mean increases from 64.33% to **69.66**% when Valve is excluded, again surpassing the best competing average (67.29%). These reference numbers are not substitutes for the official average; they merely illustrate that the gap is driven by one incompatible target rather than by systematic underperformance.

**Summary.** (i) On *domain-proximal* transfers—rotating machinery in DCASE and texture-to-texture transfers in DAGM—DAGR consistently outperforms strong baselines. (ii) On *non-proximal* transfers such as Fan→Valve, the cross-domain normal-gradient geometry is not compatible; DAGR therefore (by design) attenuates the mapped anomalous component and does not confer an advantage over target-only unsupervised methods. (iii) This behaviour delineates the *intended operating regime* of DAGR and aligns with the diagnostics introduced in the main paper. We include Valve as a negative-control case to make the boundary explicit rather than to optimise it away.

## A.4 DISCLOSURE OF LANGUAGE MODEL ASSISTANCE

Large language models were used only for editorial polishing (grammar, style, and minor rephrasing). They were not used for research design, methods, analysis, coding, figures/tables, or references. All scientific content was authored and verified by the authors, and all edits were manually reviewed. This use does not meet contributorship thresholds and does not affect reproducibility.

## A.5 COMPLEXITY AND IMPLEMENTATION

**Notation.** Let the $K$-layer source and target models have parameters $\Phi = \{\phi_1, \ldots, \phi_K\}$ and $\Psi = \{\psi_1, \ldots, \psi_K\}$. Mini-batch gradients are $\nabla\Phi$ and $\nabla\Psi$. CCCS (Sec. 3.2) yields masked gradients $\tilde{\nabla}\Phi_{\pm}$ and $\tilde{\nabla}\Psi_{+}$. ADPR (Sec. 3.3) refines a gradient variable $g_m^{(n)}$ for $n = 0, \ldots, N$ and outer step $m$, producing $\Omega_m^+ = g_m^{(N)}$ (Eq. (11)), and reuses the same map for $\Omega_m^-$, then aggregates $\Omega_m = \Omega_m^+ + \Omega_m^-$ for the target update (Eq. (3)). We follow the main text and denote the fast-weight step in Eq. (9) by $\alpha$; when disambiguation is helpful we write $\alpha_{\text{fw}}$. The CCCS masking percentile is also denoted $\alpha$ in Sec. 3.2; context makes the meaning clear.

**Per-iteration time complexity.** Let $C_{\text{fwd}}$ and $C_{\text{bwd}}$ be the cost of one forward/backward pass of the *target* network per mini-batch, and let $D$ be the number of trainable parameters.

*CCCS.* For each layer $k$ with channel set $C_k = \{c_k^1, \ldots, c_k^{P_k}\}$, CCCS computes cosine similarities $\rho_{k,p}$ between $\tilde{\nabla}\Phi_{k,p}^+$ and $\tilde{\nabla}\Psi_{k,p}^+$ and applies a percentile threshold (Eqs. (6)–(8)). All operations are on gradients already available from back-propagation; the added arithmetic is $O\left(\sum_{k,p} d_{k,p}\right) = O(D)$, where $d_{k,p}$ is the parameter count of channel $c_k^p$. This overhead is negligible compared with a single forward/backward pass.

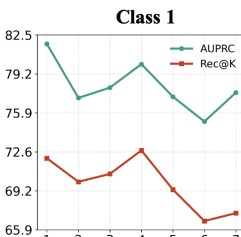 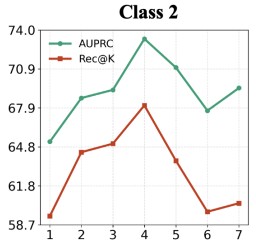 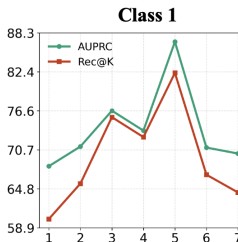

Figure 5: **Effect of channel-masking ratio $\alpha$ on DAGM transfers.** Detection performance (AUPRC and Rec@K) as $\alpha$ varies from $1\%$ to $7\%$ on Class 1, Class 3, and Class 6.

*ADPR inner loop.* At outer step $m$, set fast weights $\Psi_{m,0} = \Psi_m + \alpha\, g_m^{(0)}$ (Eq. (9)). Each inner update $g_m^{(n+1)} = g_m^{(n)} - \beta\, \nabla_g L_t\!\left(\Psi_m + \alpha\, g_m^{(n)}\right)$ (Eq. (10)) requires one forward and one backward pass through the *target* network with fast weights. Forming fast weights is an $O(D)$ axpy. With $N$ inner steps, the inner-loop cost is

$$T_{\text{ADPR}} = N\left(C_{\text{fwd}} + C_{\text{bwd}}\right) + O(D).$$

*Outer update.* A standard target update costs $(C_{\text{fwd}} + C_{\text{bwd}})$. Aggregating $\Omega_m$ (Eq. (12)) and updating $\Psi$ (Eq. (3)) adds only $O(D)$ work. Thus the per-outer-step time is

$$T_{\text{DAGR}} = (1 + N)\left(C_{\text{fwd}} + C_{\text{bwd}}\right) + O(D),$$

i.e., DAGR incurs a *constant-factor* overhead over standard training.

**Memory complexity.** Let $M_{\text{base}}$ be the peak activation/optimizer memory of the base detector. DAGR adds: (i) channel-mask buffers $\{m_{k,p}\}$ (Boolean; $\sum_k P_k$ entries), (ii) the current gradient variable $g_m^{(n)} \in \mathbb{R}^D$ and its masked initialization, and (iii) ephemeral activations for the $N$ inner steps (not accumulated across steps). No per-sample gradients nor second-order tensors are stored. Peak memory therefore satisfies

$$M_{\text{DAGR}} \approx M_{\text{base}} + O(D) + O\!\left(\sum_k P_k\right),$$

and standard techniques (gradient checkpointing, mixed precision) remain fully applicable.

**Recommended settings (used in the main experiments).** The channel-masking percentile $\alpha$ is swept from $1\%$ to $7\%$ with peaks typically in $3$–$5\%$; $\alpha = 5\%$ is adopted thereafter (Sec. 4.2, Fig. 4). The inner-loop length $N$ is kept small (a constant); the inner learning rate $\beta$ and fast-weight step $\alpha$ follow standard grids. These choices keep $T_{\text{DAGR}}$ a modest constant multiple of the baseline cost, as observed empirically.

**Stability and theory link.** Appendix A.1.2 proves that DAGR's surrogate gradient deviates from the exact target gradient by a time-independent bound, leading to inexact-descent convergence to an $O((\varepsilon + B)^2)$ neighbourhood without error accumulation; this matches the smooth training curves observed across tasks.

**Practical deployment note (non-parametric).** When transfers involve potentially non-adjacent domains, practitioners may *screen* the proximity of *normal* gradient distributions (e.g., via $W_1$ or MMD) before enabling gradient reuse. If the divergence is large, a conservative fallback is to disable reuse and proceed with target-only updates. This screening is advisory and does not alter the reported experiments; devising automated reliability weighting is left for future work.

# B  EFFECT OF CHANNEL-MASKING RATIO ON DAGM BENCHMARKS

To examine whether the channel-masking ratio $\alpha$ identified on the DCASE benchmarks is also reasonable for visual-texture domains, we conduct an additional ablation study on the DAGM transfers (Class 2 $\to$ Class 1/3/6, as shown in Figure 5). For each target class, $\alpha$ is swept from $1\%$ to $7\%$, and we report AUPRC and Rec@K averaged over three runs.

**Results.**   The three DAGM targets exhibit heterogeneous behaviours. One class shows a mostly decreasing trend as $\alpha$ grows, whereas the other two classes first benefit from masking a small fraction of channels and then degrade when too many channels are removed. Across all cases, however, the $\alpha \in [3\%, 5\%]$ range consistently yields competitive performance: the scores at $\alpha = 5\%$ are either close to the per-class maximum (within a small margin) or lie on a relatively flat part of the curve without sharp deterioration. This indicates that the method is not overly sensitive to the exact choice of $\alpha$ in this moderate region, even though the precise optimum is slightly domain dependent.

**Conclusion.**   Taken together with the DCASE study in the main paper, these experiments suggest that $\alpha = 5\%$ is a *robust default* that performs reasonably well across both acoustic (DCASE) and visual-texture (DAGM) benchmarks, rather than a strictly optimal setting for every single transfer. In practice, mild per-domain tuning around this range could further improve performance if desired, but all reported results use the same fixed value $\alpha = 5\%$ for simplicity and fairness.

