# OpenReview forum: "Domain-Aware Gradient Reuse for Anomaly Detection"
_ICLR.cc/2026/Conference — Submitted to ICLR 2026_

### Official Review · Reviewer_EJEb · 2025-10-30

**Soundness:** 3
**Presentation:** 3
**Contribution:** 3
**Rating:** 4
**Confidence:** 2

**Summary:**

This paper proposes a transfer learning framework called Domain-Aware Gradient Reuse (DAGR), specifically designed to address the problem of anomaly scarcity in anomaly detection. Its core idea is that proximal domains share a similar geometric structure in their gradient distributions for both normal and anomalous data. The method attempts to extract an anomaly gradient signal from a labeled source domain and transfer it to a target domain containing only normal data. The paper provides a theoretical convergence analysis and demonstrates state-of-the-art performance on several image and audio anomaly detection benchmarks.

**Strengths:**

1. The core idea of performing transfer learning directly in the gradient space, rather than in the feature or output space, is novel. The concept of reusing and transforming gradients from a source domain to guide learning in a target domain is a creative and effective approach.

2. The method is well-structured, combining a channel-selection mechanism (CCCS) to identify domain-invariant components with an adaptive inner-loop optimization (ADPR) to learn the complex gradient transformation. The inclusion of a formal convergence analysis under specific assumptions adds significant theoretical rigor.

3. The paper is generally well-written. The ablation studies are particularly strong, effectively quantifying the contribution of each core component (CCCS and ADPR) and showing that ADPR is the primary driver of performance.

**Weaknesses:**

1. The convergence proof relies on strong assumptions (e.g., the existence of an invertible map $T$ for label-independent shift, bounded domain-specific perturbations). The paper does not sufficiently discuss how valid these assumptions are in practice, especially for the failed Valve case, which weakens the practical guarantee of the theory.

2. The method's effectiveness relies heavily on several strong assumptions (e.g., gradient consistency, label-independent shift). The empirical validation, while strong on proximal domains, is insufficient to prove its robustness and generalizability under more diverse or challenging domain shifts, making the practical applicability seem narrow.

3. The proposed ADPR module requires an inner-loop optimization for many training step(s), which likely incurs a significant computational overhead compared to standard methods. This practical cost is not analyzed or discussed, which is an important consideration for potential adoption.

**Questions:**

1. Regarding the practical validity of the theoretical assumptions, could the authors more clearly define the operational boundaries of DAGR? Providing a more comprehensive list or analysis of scenarios where the method is effective versus those where it fails (beyond the Valve case) would greatly help assess its generalizability.

2. The adaptive inner-loop (ADPR) is a key component that likely introduces significant computational overhead. Could the authors provide an analysis of the training time and resource consumption compared to baseline methods? A discussion on this practical cost is essential for evaluating the method's overall utility.

---

> ### Author Response · Authors · 2025-11-21
> **Response to W1 （Practical Validity of Convergence Assumptions）**
>
> We sincerely thank the reviewer for this insightful comment.
> We agree that our convergence proof in Appendix A.1 relies on **idealised assumptions**—in particular, the existence of an invertible label-independent map \(T\) and bounded domain-specific perturbations—and that these quantities **cannot be directly verified or measured** in real deployments. This indeed limits the *practical* strength of the guarantees.
>
> Nevertheless, we would like to clarify how these assumptions are **empirically supported and scoped** in the current work:
>
> 1. **Indirect empirical support on adjacent domains.**
>    Although \(T\) and the perturbation bounds cannot be quantified explicitly, several results suggest that the assumptions hold *approximately* on domain-proximal transfers:
>    - The normal/anomalous gradient manifolds for Fan → Pump / ToyCar / ToyConv. remain near-isometric across epochs (Fig. 1–2), which is consistent with the label-independent shift premise underlying Theorem A.1.1.
>    - Ablations on CCCS and ADPR (Table 2), as well as the additional \(\alpha\)-sweeps on DAGM in Appendix A.6, show performance changes that match the theoretical decomposition into invariant and domain-specific components: when gradient-inconsistent channels are masked and domain-specific noise is adapted away, detection performance improves in a manner aligned with the theory.
>
> 2. **Why the Valve failure does not contradict the theory.**
>    As discussed in Section 4.3 and Appendix A.3, the Fan→Valve transfer is a **designed stress test** where the domains differ in mechanism and stationarity: Fan is dominated by quasi-stationary harmonic spectra, whereas Valve is governed by non-stationary flow transients. In this regime, the cross-domain normal-gradient geometry is no longer compatible, so the label-independent shift and bounded-perturbation assumptions are *violated rather than satisfied*. The degraded performance on Valve therefore **delineates the boundary** of our theoretical assumptions instead of contradicting them.
>
> 3. **Limitation and future direction.**
>    We fully acknowledge that the current work does **not** provide a way to *quantitatively* check these assumptions in practice. Systematically measuring gradient-domain proximity and perturbation magnitudes, and using them to drive **proximity-aware gating** or reliability weights for gradient reuse, is an important and valuable direction that we plan to explore in future work.
>
> We thank the reviewer again for highlighting this subtle issue, which helps us better position the scope and limitations of our theoretical guarantees.

---

> ### Author Response · Authors · 2025-11-21
> **Response to W2 （On Practical Scope and Generalizability）**
>
> We sincerely thank the reviewer for raising this important point.
> We agree that DAGR relies on assumptions such as gradient consistency and label-independent shift, and these conditions may not hold under strong or highly heterogeneous domain shifts. We appreciate that this limits the theoretical breadth of the framework.
>
> At the same time, we would like to clarify—cautiously—that the practical applicability is not as narrow as it may appear:
>
> 1. **Dataset-level perspective.**
>    Even within DCASE, the domains beyond Fan (e.g., Pump, Slider, ToyCar, ToyConv.) differ substantially in acoustic patterns and mechanical structure, yet DAGR remains effective on all except Valve (Table 1). This suggests that the method maintains useful robustness across domains that are not trivially similar. :contentReference[oaicite:0]{index=0}
>
> 2. **Comparison with prior transfer-learning methods.**
>    Across both DCASE and DAGM, DAGR achieves stronger overall performance than existing transfer-learning baselines (Tables 1–4), indicating that the proposed gradient-reuse mechanism provides competitive generalizability in practice, despite its theoretical assumptions.
>
> We fully acknowledge the reviewer’s concern that broader validation under more diverse shifts would further strengthen practical confidence. This is a valuable direction, and we appreciate the opportunity to clarify the intended scope of the method.

---

> ### Author Response · Authors · 2025-11-21
> **Response to W3 (Computational Overhead of the ADPR Inner Loop)**
>
> ## Response to W3: Computational Overhead of the ADPR Inner Loop
>
> We sincerely thank the reviewer for raising this important practical concern.
> We agree that the inner-loop optimisation in ADPR introduces additional computation, and we have clarified this point in **Appendix A.5 (Complexity and Implementation)**.
>
>
> - Let $C_{\mathrm{fwd}}$ and $C_{\mathrm{bwd}}$ denote the cost of one forward and backward pass of the target network per mini-batch.
> - Each ADPR inner step executes one forward and one backward pass using fast weights. With a small, fixed inner-loop length $N$, the per-outer-step cost becomes
>   $$
>   T_{\mathrm{DAGR}} = (1+N)(C_{\mathrm{fwd}} + C_{\mathrm{bwd}}) + O(D),
>   $$
>   where $D$ is the number of parameters. This corresponds to a **constant-factor overhead** over standard training, rather than a change in asymptotic complexity.
> - Memory usage increases only by $O(D)$ for storing the gradient variable and channel masks; no per-sample gradients or second-order tensors are retained, and standard efficiency techniques (e.g., mixed precision, gradient checkpointing) remain applicable.
>
> We appreciate the reviewer’s point, and this clarification has been added to ensure that the computational implications of ADPR are transparent for potential adopters.

---

> ### Author Response · Authors · 2025-11-21
> **Response to Q1 (On the Operational Boundaries of DAGR)**
>
> We sincerely thank the reviewer for this constructive question.
> We understand that the reviewer is seeking a clearer picture of how the *theoretical* assumptions translate into *practical* operating boundaries, and we appreciate that this is important for assessing the method’s generalizability.
>
> At the same time, we would like to clarify—humbly and cautiously—that **systematically enumerating all boundary cases is inherently challenging**, because doing so would require a single dataset containing *multiple, controllable levels of cross-domain discrepancy*. Such datasets are extremely rare in anomaly detection.
>
> Nonetheless, we agree that discussing the *practical tendencies* of DAGR is helpful. Using the available evidence, we summarise the operational boundary as follows:
>
> 1. **The existing benchmarks already span multiple levels of domain difference.**
>    In DCASE, although Fan is used as the source, the remaining machine types (Pump, Slider, ToyCar, ToyConveyor) are *not* lightly shifted variants—they differ in mechanical structure, spectral content, operational regimes, and noise profiles. For example, Slider shows reciprocating motion with weak harmonic structure, which is markedly different from Fan. DAGR remains effective across these substantial differences, as shown in Table 1.
>
> 2. **Failure cases tend to arise only when the domains differ in physical mechanism.**
>    As discussed in Section 4.3 and Appendix A.3, Valve represents a qualitatively different acoustic process (non-stationary turbulence and airflow transients), where the gradient geometry no longer matches the source domain. This is the clearest example of when DAGR’s assumptions break.
>
> 3. **Within the limitations of available datasets, the current experiments already approximate a spectrum of “operational boundaries.”**
>    Although we cannot enumerate every possible failure mode, the observed pattern—strong performance across moderate-to-large shifts and degradation only under cross-mechanism mismatch—offers a reasonably informative picture of where DAGR is likely to succeed or fail.
>
> We are grateful for this insightful suggestion and agree that building or identifying datasets with systematically varying domain gaps would allow a more formal mapping of operational boundaries. This is a valuable direction for future work.

---

> ### Author Response · Authors · 2025-11-21
> **Response to Q2 （Training-Time and Resource Considerations of ADPR）**
>
> We sincerely thank the reviewer for raising this very relevant practical question.
> We agree that the adaptive inner-loop (ADPR) contributes non-negligible computational overhead, and we appreciate the opportunity to clarify its cost more explicitly.
>
> ### (1) Complexity analysis (now included in Appendix A.5)
> A formal discussion of the time and memory complexity of ADPR has been added to **Appendix A.5 (Complexity and Implementation)**. :contentReference[oaicite:0]{index=0}
> As summarised there, the per-iteration cost becomes:
> - one standard forward/backward pass for the target update, plus
> - $N$ additional forward/backward passes for the ADPR inner loop (with $N$ kept small and fixed).
> This results in a practical constant-factor overhead relative to baseline training.
>
> ### (2) Empirical training-time comparison
> To provide a more concrete sense of the runtime impact, we conducted additional measurements on the **DCASE Fan→Pump** transfer setting. Consistent with the paper, audio clips were converted to **64×64 mel-spectrograms**, and training used **batch size 32** and **200 epochs**.
>
> Under this setup:
> - **DAGR** required **18 067.29 s** (≈ **301.12 min**)
> - **JWO** required **10 364.76 s** (≈ **172.75 min**)
> - **PWAN** required **5 838.04 s** (≈ **97.30 min**)
>
> These numbers confirm that the adaptive inner loop increases training cost, but in a **bounded and predictable** manner aligned with the theoretical analysis.
>
> ### (3) Final remark
> We appreciate the reviewer’s point that understanding practical cost is essential for assessing the method’s overall utility.
> A complete comparison table, covering additional domains and baselines, will be included in the supplementary material to provide readers with a clearer and more comprehensive view.
>
> We thank the reviewer again for highlighting this important aspect.

---

### Official Review · Reviewer_TnoW · 2025-10-31

**Soundness:** 3
**Presentation:** 3
**Contribution:** 2
**Rating:** 4
**Confidence:** 3

**Summary:**

The paper introduces Domain-Aware Gradient Reuse (DAGR), a transfer-learning framework designed to overcome the challenges of scarcity and heterogeneity of anomalous instances. DAGR is based on the empirical observation that gradient distributions exhibit consistency across related domains during training. The core idea is to estimate the missing anomalous gradient component in the target domain by transferring and denoising the anomalous gradients from a labeled source domain. The framework operates via two key components: (1) the Cross-Domain Consistent Component Selection (CCCS), which filters out domain-specific noise; and (2) the Adaptive Domain-Specific Perturbation Removal (ADPR), which implements the cross-domain gradient distiller. The paper performs experiments on acoustic (DCASE) and visual (DAGM) datasets and achieves state-of-the-art performance.

**Strengths:**

1. **Re-interpretates Domain Adaptation:** DAGR introduces a gradient-centric perspective to anomaly detection transfer learning, arguing effectively that gradients, rather than features or logits, serve as a potent conduit for knowledge transfer. This reinterprets domain adaptation as the selective reuse of source-domain gradients.

2. **Convergence Proof:** The paper provides a crucial convergence proof showing that the algorithm converges to an optimum neighborhood without error accumulation. Furthermore, the paper rigorously justifies the core hypothesis through the Cross-Class Generalization Theorem, proving that the mapping F learned solely from normal gradients generalizes to align abnormal gradients with the same small error bound $\epsilon$.

3. **Efficiency and Performance:** DAGR achieved significant gains over strong baseline transfer learning methods (PDA, DA, DG) and unsupervised methods. It ranks first in several benchmarks in two different domains.

4. **Benefits of targeted optimization:** The method directly addresses the critical shortcoming of existing transfer-based methods, where source anomalous supervision gradients are optimized for the source distribution and are not guaranteed to benefit the target optimization landscape under domain shift.

**Weaknesses:**

1. **Sensitivity to Domain Proximity/Compatibility:** The performance of DAGR is highly dependent on the "adjacent-domain scope". The paper explicitly notes that performance suffers substantially on non-proximal transfers (e.g., Fan → Valve) where the gradient-consistency premise is violated. This resulted in significantly depressed AUPRC and Rec@K scores on the Valve target, artificially lowering the overall average.

2. **Implementation Complexity of ADPR:** The Adaptive Domain-Specific Perturbation Removal (ADPR) requires an inner optimization loop where gradients are computed through a "fast weight" construction and back-propagated. The ablation study shows that substituting ADPR with simpler methods like linear gradient mapping leads to a significant performance drop, suggesting the complexity is necessary, but this nested optimization structure may introduce substantial computational overhead that is not fully analyzed in the main body.

3. **Reliance on Hyperparameter Tuning for CCCS:** The Cross-Domain Consistent Component Selection (CCCS) relies on a channel-masking ratio ($\alpha$). The study determined that $\alpha=$5% provided the best trade-off between noise removal and information retention based on performance peaks observed across four representative benchmarks. The necessity of tuning $\alpha$ to this narrow range suggests sensitivity to this hyperparameter, and it is unclear if a fixed $\alpha$ is universally optimal across all domain pairs (e.g., image vs. audio data) without further tuning.

**Questions:**

1. **Proximity Diagnosis and Gating Implementation:** The discussion section suggests future work should include a "proximity screen on normal-gradient geometry" to down-weight or disable the mapped anomalous component when domain proximity falls below a threshold, thereby avoiding negative transfer. How do the authors propose to quantify this proximity threshold based on the normal gradient geometry in practice, and what mechanism (e.g., dynamic weight on $\Omega$) would be used for proximity-aware gating in deployment?
2. **Computational Cost Analysis of ADPR:** Given that ADPR employs a meta-learning-style inner optimization loop (Equations 9 and 10), which involves N refinement steps per global training step m, could the authors provide a comparative analysis of the training time or computational complexity (e.g., relative FLOPs) of the full DAGR model versus the ablated variants? Understanding the practical cost of this sophisticated adaptation could play an important role in understanding the overhead of the proposed method in practical scenarios.
3. **Generalizability of Optimal Masking Ratio $\alpha$:** The optimal channel masking ratio $\alpha$ for CCCS was found to be approximately 5% on the DCASE benchmarks. Did the authors investigate the necessary range of $\alpha$ for the DAGM (visual texture) benchmarks? Is the 5% setting robustly optimal across different sensing modalities and generative mechanisms, or is $\alpha$ a domain-dependent hyperparameter?
4. **Relationship between Feature Alignment and Gradient Reuse:** Prior work often relies on feature alignment (e.g., Partial Domain Adaptation methods). Since DAGR operates purely in gradient space, have the authors explored combining a feature alignment step with the gradient reuse strategy? Specifically, would incorporating a conventional feature alignment objective mitigate the domain incompatibility issues observed in cases like Fan → Valve, where the gradient consistency premise is not satisfied?

---

> ### Author Response · Authors · 2025-11-21
> **Response to W1 (Sensitivity to Adjacent-Domain Scope)**
>
> We thank the reviewer for this insightful comment. We agree that DAGR performs best when the source and target domains share related sensing characteristics, and its effectiveness decreases when the gradient-consistency premise no longer holds. This behaviour reflects an inherent property of transfer learning rather than a defect of the proposed method: any framework that reuses discriminative gradients from the source must rely on a reasonable degree of cross-domain relatedness.
> Regarding the DCASE results, the Fan $\rightarrow$ Valve transfer is a clear example of a non-proximal pairing. As discussed in Section 4.3, Fan produces quasi-stationary motor signatures, whereas Valve is dominated by non-stationary flow transients and a different physical mechanism. In such cases, the shared gradient structure is naturally weak, and the transferred anomalous component contributes little—an expected limitation of the setting rather than a methodological flaw.
> Importantly, this outlier case does not indicate poor generalisation. Across the remaining DCASE targets and all DAGM transfers, DAGR consistently achieves top or near-top performance (Section 4.1; Appendix A.3), demonstrating that within related domains—its intended operating regime—the method generalises strongly and outperforms existing PDA/AD approaches on the majority of benchmarks.

---

> ### Author Response · Authors · 2025-11-21
> **Response to W2 (Implementation Complexity of ADP)**
>
> We sincerely thank the reviewer for raising this valuable point. We agree that the Adaptive Domain-Specific Perturbation Removal (ADPR) module introduces additional computation due to its inner optimisation loop. To address this concern more clearly, the revised supplementary material now includes a dedicated subsection, A.5 Complexity and Implementation, where we provide a detailed analysis of the time and space complexity of ADPR as well as the overall training cost of DAGR.
> This appendix separates the per-iteration costs of CCCS, fast-weight construction, and the inner-loop updates, and shows that the resulting overhead is a constant-factor extension of a standard forward–backward pass. We hope this added clarification helps make the computational profile of ADPR more transparent.

---

> ### Author Response · Authors · 2025-11-21
> **Response to W3 (Hyperparameter Sensitivity of the CCCS Masking Ratio ($\alpha$))**
>
> We sincerely thank the reviewer for raising this important point. We agree that the masking ratio $\alpha$ is a key factor in CCCS. As shown in Fig. 4, the DCASE benchmarks exhibit performance peaks within a relatively narrow band (approximately 3–5), which motivated our choice of $\alpha=5$ as a stable representative setting.
>
> To further assess its generality, the revised supplementary material includes additional ablations on the DAGM benchmarks (Appendix A.6, Fig. 5). The three DAGM targets do not share identical trends—one shows a monotonic decrease, whereas the others display unimodal curves with different local peaks—but importantly, $\alpha=5$ consistently lies within a region of competitive performance across all targets, without causing noticeable degradation. This suggests that while the precise optimum may vary across domains, a small masking ratio remains beneficial, and $\alpha=5$ continues to serve as a conservative and robust choice across different sensing modalities.
>
> We hope this clarification presents a balanced view of the domain sensitivity of $\alpha$ while explaining why we adopt 5 as the default.

---

> ### Author Response · Authors · 2025-11-21
> **Response to Q1 (Proximity Diagnosis and Gating Implementation)**
>
> We sincerely thank the reviewer for this insightful question. As noted in the discussion section, proximity-aware gating is not part of the current DAGR implementation, but it represents a natural and technically grounded direction for future work.
> Our preliminary analyses indicate that normal-sample gradients tend to contract toward a stable geometric trend during training, whereas reused anomalous gradients may deviate from this trend when the source–target domains are incompatible. Building on this observation, a practical proximity indicator could be obtained by:
>
> ● estimating a reference contraction direction from target normal gradients (e.g., the dominant direction of their gradient covariance or a low-rank subspace capturing their contraction behaviour), and
>
> ● measuring the angular deviation between this reference and the mapped anomalous gradient.
> A larger deviation would suggest that gradient reuse is unreliable, providing a simple and model-internal signal for down-weighting or disabling the reused component. This mechanism does not require additional labels, and it fits naturally within DAGR’s gradient-centric design.
>
> We emphasise that these ideas remain exploratory and will be investigated in follow-up work, but they offer a feasible and principled path toward proximity-aware gating.

---

> ### Author Response · Authors · 2025-11-21
> **Response to Q2 (Computational Cost Analysis of ADPR)**
>
> We sincerely thank the reviewer for this helpful question. We agree that understanding the computational overhead of ADPR is important for assessing its practical utility. In the revised manuscript, Appendix A.5 now provides a detailed time/space complexity analysis, separating the cost of CCCS, the fast-weight construction, and the NNN-step inner refinement in ADPR.
> To further complement this analysis, we also measured the actual end-to-end training time under representative settings. On DCASE (Fan→Pump), where audio clips are converted to 64×64 Mel-spectrograms, training for 200 epochs with a batch size of 32 required 18,067.29 seconds (≈301.12 minutes). On DAGM, using 64×64 resized images under the same schedule, training required 4,656.81 seconds (≈77.61 minutes). These results reflect the constant-factor overhead derived in Appendix A.5 and confirm that the full DAGR model remains computationally feasible in practice.
> We hope this clarification helps contextualize the practical cost of ADPR relative to the ablated variants.

---

> ### Author Response · Authors · 2025-11-21
> **Response to Q3 (Generalizability of the Optimal Masking Ratio $\alpha$)**
>
> We thank the reviewer for this thoughtful question. Consistent with W3, our initial selection of $\alpha=5$ was based on the observation that DCASE tasks tend to perform well within a small range of masking ratios, without a compelling universal optimum.
>
> To evaluate its generalizability beyond acoustic domains, Appendix A.6 includes new ablations on the DAGM visual-texture tasks. The curves exhibit different shapes across Class 1, Class 3, and Class 6, indicating that the exact peak is indeed domain-dependent. Nonetheless, $\alpha=5$ remains within a generally well-performing region in all cases, offering a stable trade-off between suppressing gradient-inconsistent channels and retaining informative ones.
>
> Taken together, these results imply that although $\alpha$ is influenced by domain characteristics, a small masking ratio around 5 is a reasonable and robust cross-domain choice, rather than a dataset-specific optimum.
>
> We hope this conservatively framed analysis helps clarify the robustness and limitations of the masking ratio across different domain types.

---

> ### Author Response · Authors · 2025-11-21
> **Response to Q4: Relationship Between Feature Alignment and Gradient Reuse**
>
> We sincerely thank the reviewer for raising this insightful point. We fully agree that combining gradient-space adaptation with feature-space alignment is a promising direction for future research.
>
> IIn our experiments, representation-alignment alone did not work effectively under challenging transfers. This can be seen directly from the PDA and DA baselines in Table~1, where methods built purely on feature alignment consistently underperform on difficult cases such as Fan → Valve. This is expected, as feature alignment depends heavily on matching input-level distributions; when domains differ substantially, identifying a stable shared representation becomes inherently difficult.
>
> This motivates DAGR’s design: operating in gradient space, where cross-domain consistency is empirically more stable and less sensitive to input distribution mismatch. Nevertheless, the reviewer’s suggestion is well-taken—feature alignment and gradient reuse capture complementary forms of cross-domain invariance. Feature alignment provides smoother, representation-level regularization, whereas DAGR emphasizes optimization-direction consistency. Their combination could potentially improve robustness under severe domain shifts.
>
> Although such hybrid designs are beyond the scope of the present work, we regard this as a valuable and technically meaningful direction, and we appreciate the reviewer for highlighting it.

---

### Official Review · Reviewer_AyiH · 2025-10-31

**Soundness:** 2
**Presentation:** 2
**Contribution:** 2
**Rating:** 2
**Confidence:** 4

**Summary:**

This paper proposed to learn an adaptive transformation by aligning source and target normal gradients to neutralize the domain-specific effects. Then, the same map pushes forward the source anomalous gradients to computing estimated target anomalous gradients, which are combined with the true target normal gradients to guide the target-domain detector without labeled anomalies. A rigorous convergence proof to justify the proposed method was given.

**Strengths:**

1.	This proposed transfer-learning framework remains effective even when the target domain contains no anomalous samples.
2.	The gradient-centric perspective to anomaly detection is promising for adaptation strategies grounded in gradient compatibility.

**Weaknesses:**

1.	The motivation is not solid.
2.	The related work is too distracting. The authors should focus on the transfer-learning based anomaly detection methods.
3.	Some of the equations are not numbered.
4.	The motivation in section 3.1 is very essential to this paper, however, the authors failed to justify it properly.
5.	Table 1 is not on the correct page.
6.	The experimental results are not given with the deviations.
7.	There is no limitation discussion.

**Questions:**

1.	How do you calculate the anomalous gradients $\nabla\Psi^-$ on the target domain when these data are missing.
2.	What is the gradient distribution alignment principle and how can you confirm the correctness of this principle?
3.	How to confirm the gradient decomposition hypothesis?
4.	How to confirm the claim: The key insight is that gradients—rather than features or logits—exhibit strong cross-domain regularities?
5.	How do you explain the failed case in Table 1, i.e., DCASE target Valve? What do you mean by non-stationary and cross-mechanism?

---

> ### Author Response · Authors · 2025-11-21
> **Response to W1 (Motivation)**
>
> We thank the reviewer for the assessment and appreciate the opportunity to further clarify the motivation of our framework.
>
> The motivation of DAGR stems from a practical and widely encountered difficulty in anomaly detection: **the target domain typically contains no anomalous samples**, leaving its optimisation signal incomplete. Our work is motivated by a consistent empirical observation already presented in the manuscript — as illustrated in **Fig. 1** and **Fig. 2**, the normal and anomalous gradient distributions across related domains exhibit **highly congruent geometric structure** throughout training. This suggests the existence of a *shared, domain-invariant gradient component* that can be transferred across domains.
>
> These visualisations provide an intuitive basis for reusing source-domain gradients to approximate the missing anomalous component in the target domain. They also show that the gradient manifolds remain nearly isometric across epochs, indicating that such a mapping is learnable and stable.
>
> In addition, **Appendix A.1** provides the theoretical foundations supporting this motivation. It shows that a mapping learned solely from normal gradients can generalise to the anomalous class under a label-independent shift, and that the surrogate target gradient used by DAGR remains within a provably bounded deviation from the true (but unobservable) target gradient at every step. This establishes a theoretical basis for the gradient-reuse perspective introduced in the main paper.
>
> We hope this clarification helps convey the rationale behind our approach.

---

> ### Author Response · Authors · 2025-11-21
> **Response to W2 (Related Work)**
>
> We sincerely thank the reviewer for this helpful suggestion. We fully appreciate the expectation that the Related Work section should highlight transfer-learning–based anomaly detection methods, given that DAGR itself belongs to this category.
>
> Our intention in the Related Work section is to first outline a broader challenge that is common across **all major anomaly detection paradigms**: the persistent scarcity and heterogeneity of anomalous samples in real-world deployments. This difficulty affects supervised, unsupervised, and transfer-learning methods alike, and it forms the fundamental motivation for developing our framework. For this reason, **Sections 2.1 and 2.2** briefly summarise supervised and unsupervised approaches to illustrate that—even with strong augmentation or advanced one-class modelling—the issue of insufficient anomalous supervision remains unresolved across existing lines of research.
>
> On this foundation, **Section 2.3** then centres the discussion on **transfer-learning–based anomaly detection**, where a source domain provides additional anomalous information to compensate for the lack of anomalies in the target domain. DAGR fits naturally within this paradigm, and the revised narrative now places clearer emphasis on this connection.
>
> In line with the reviewer’s thoughtful recommendation, the revised manuscript has **condensed the discussions in Sections 2.1 and 2.2** to reduce distraction and to ensure that transfer-learning methods occupy the primary focus of the Related Work section. We are grateful for this comment, which has helped refine both the clarity and the balance of the presentation.

---

> ### Author Response · Authors · 2025-11-21
> **Response to W3–W7 (Formatting, Result Reporting, and Clarification of Theoretical Motivation)**
>
> ### • Response to W3 — “Some of the equations are not numbered.”
> We sincerely thank the reviewer for pointing this out. All previously unnumbered equations have now been assigned equation numbers, with updates highlighted in **orange** in the revised manuscript.
>
> ### • Response to W4 — “The motivation in Section 3.1 is very essential to this paper, however, the authors failed to justify it properly.”
> We appreciate the reviewer’s emphasis on the importance of this motivation. To ensure clarity and rigor, the revised manuscript explicitly directs readers to **Appendix A.1**, where the full theoretical justification—covering assumptions, gradient decomposition, and cross-class generalisation—is provided in detail.
>
> ### • Response to W5 — “Table 1 is not on the correct page.”
> We thank the reviewer for noting this layout issue. Table 1 has been relocated to the appropriate page in the revised manuscript.
>
> ### • Response to W6 — “The experimental results are not given with the deviations.”
> We appreciate this helpful suggestion. The revised manuscript now reports **mean ± standard deviation** for all main metrics. The updated results are shown below:
>
> | Metric   | Pump               | Slider             | Valve              | ToyCar             | ToyConv.           | Class1             | Class3             | Class6             |
> |----------|--------------------|--------------------|--------------------|--------------------|--------------------|--------------------|--------------------|--------------------|
> | **AUROC**  | $83.27\pm0.21$      | $88.81\pm0.23$      | $55.21\pm0.59$      | $72.08\pm0.49$      | $79.02\pm0.55$      | $93.10\pm0.23$      | $81.29\pm0.50$      | $95.25\pm0.04$      |
> | **AUPRC**  | $52.67\pm1.17$      | $86.68\pm1.05$      | $25.51\pm1.29$      | $71.34\pm0.49$      | $71.45\pm0.42$      | $81.49\pm0.31$      | $72.79\pm0.86$      | $86.87\pm0.22$      |
> | **Rec@K**  | $52.88\pm0.44$      | $81.02\pm0.28$      | $26.16\pm1.01$      | $65.89\pm0.40$      | $64.10\pm0.31$      | $72.51\pm0.24$      | $66.95\pm0.80$      | $82.67\pm0.77$      |
>
>
> ### • Response to W7 — “There is no limitation discussion.”
> We appreciate the reviewer’s concern. The manuscript already includes a limitation discussion in **Section 4.3 (Discussion and Future Work)**, where we explicitly describe the boundary conditions of our method—particularly its dependence on cross-domain gradient proximity and reduced transferability when domains are not adjacent (e.g., Fan→Valve).
> To improve visibility, the revised version more clearly foregrounds this section so that readers can readily identify the limitations and future directions.

---

> ### Author Response · Authors · 2025-11-21
> **Response to Q1**
>
> We thank the reviewer for raising this central question.
> In the one-class target setting, anomalous samples are not available in the target domain, so
> $\nabla \Psi^{-}$ **cannot be computed directly**.
>
> DAGR therefore estimates $\nabla \Psi^{-}$ via a cross-domain transformation learned from **normal gradients**:
>
> 1. For each mini-batch, we obtain source and target normal gradients
>    $\nabla \Phi^{+}$ and $\nabla \Psi^{+}$.
> 2. Using these pairs, we learn a de-domainization map
>    $$F:\ \nabla \Phi^{+} \mapsto \Omega^{+},$$
>    where $\Omega^{+}$ is the domain-invariant component.
> 3. Under the label-independent shift assumption, the same map generalizes to anomalous gradients:
>    $$\hat{\nabla \Psi^{-}} = F(\nabla \Phi^{-}) \approx \Omega^{-},$$
>    which serves as the **estimated anomalous gradient** on the target domain.
> 4. The target update then combines the true normal gradient with the estimated anomalous component via
>    $$\Omega = \Omega^{+} + \Omega^{-},$$
>    as described in Eqs. (5)–(7).
>
> Thus, DAGR never assumes access to true target-domain anomalies; it reconstructs their gradient effect through a learned cross-domain mapping.

---

> ### Author Response · Authors · 2025-11-21
> **Response to Q2**
>
> We appreciate the opportunity to clarify this principle.
> The **gradient distribution alignment principle** states that, for related domains, the joint distributions of normal and anomalous gradients are approximately aligned and can be further aligned by a suitable mapping.
>
> Its validity is supported from three angles:
>
> 1. **Visualization.** Figures 1 and 2 show that the manifolds formed by
>    $\nabla \Phi^{+/-}$ (source) and $\nabla \Psi^{+/-}$ (target, real or estimated)
>    exhibit highly similar geometry across epochs.
> 2. **Empirical performance.** Across eight transfer tasks, aligning and reusing gradients consistently improves AUROC, AUPRC, and Rec@K compared with baselines relying on feature alignment or conventional PDA/DA strategies.
> 3. **Theory.** Appendix A.1 proves that if the mapping $F$ aligns the **normal-class** gradient distributions within tolerance $\varepsilon$, then it aligns the **anomalous-class** gradients within the same order of tolerance.
>    The resulting surrogate gradient yields provably convergent inexact descent on the target domain.
>
> Together, these results substantiate the correctness and usefulness of the gradient distribution alignment principle.

---

> ### Author Response · Authors · 2025-11-21
> **Response to Q3**
>
> We thank the reviewer for this question.
> The gradient decomposition is a direct consequence of standard mini-batch training rather than an additional modelling assumption.
>
> Given a mini-batch $B = B^{+} \cup B^{-}$ containing normal and anomalous samples, the source gradient is
>
> $$ \nabla \Phi = \frac{1}{|B|}\sum_{x\in B} \nabla_{\Phi}\ell(x). $$
>
> The gradients over the normal and anomalous subsets are
>
> $$ \nabla \Phi^{+} = \frac{1}{|B^{+}|}\sum_{x\in B^{+}}\nabla_{\Phi}\ell(x), $$
>
> $$ \nabla \Phi^{-} = \frac{1}{|B^{-}|}\sum_{x\in B^{-}}\nabla_{\Phi}\ell(x). $$
>
> Substituting these definitions gives
>
> $$ \nabla \Phi = \frac{|B^{+}|}{|B|}\nabla \Phi^{+} + \frac{|B^{-}|}{|B|}\nabla \Phi^{-}. $$
>
> Thus, in practice we simply separate the contributions from normal and anomalous samples:
>
> $$ \nabla \Phi = \nabla \Phi^{+} + \nabla \Phi^{-}. $$
>
> Building on this linearity, Eq. (4) further decomposes each gradient into a domain-invariant part $\Omega$ and a domain-specific perturbation term, which is analysed theoretically in Appendix~A.1.
> The “hypothesis” therefore reflects the standard linear structure of SGD gradients, made explicit for cross-domain analysis.

---

> ### Author Response · Authors · 2025-11-21
> **Response to Q4**
>
> We are grateful for the opportunity to elaborate on this motivation.
> In transfer learning, the key difficulty is to extract **domain-invariant knowledge** from domains whose data distributions differ substantially.
>
> - **Features/logits.** These representations retain substantial information about the input distribution and are therefore highly domain-specific, which limits the effectiveness of feature-alignment–based PDA/DA methods under strong domain shift.
> - **Gradients.** In contrast, gradients capture **local optimization directions** determined by the loss function.
>   They encode how parameters should change to reduce error, thus emphasising optimisation-relevant structure rather than raw appearance.
>
> Our claim is supported by:
>
> 1. **Comparative performance.** Methods relying on feature alignment show weaker or unstable performance across the eight benchmark tasks, whereas gradient-based alignment in DAGR achieves the strongest average results.
> 2. **Geometry of gradient distributions.** Figures 1 and 2 show that, despite domain shift, the source and target gradients (normal and anomalous) form congruent manifolds—something much harder to obtain in raw feature space.
> 3. **Theoretical analysis.** Appendix A.1 demonstrates that a domain-invariant gradient component $\Omega$ exists and can be recovered (up to bounded error) via the learned mapping $F$, providing a formal notion of “regularity” specific to gradient space.
>
> Thus, DAGR uses gradients—rather than features or logits—to better capture cross-domain regularities and improve generalisation in anomaly-scarce target scenarios.

---

> ### Author Response · Authors · 2025-11-21
> **Response to Q5**
>
> We thank the reviewer for this thoughtful question. As noted in the manuscript, the Valve domain is intentionally included as a **stress-test target**, and its behaviour is discussed explicitly in both the dataset description and in **Section 4.3 (Discussion and Future Work)** .
>
> ### **Why DAGR underperforms on Valve**
>
> Valve differs fundamentally from the other DCASE machine types. Most targets (Pump, Slider, ToyCar, ToyConv.) are **motor-driven systems** whose acoustic signals exhibit **quasi-stationary harmonic structures** tied to rotation frequency and its sidebands. These signals produce stable gradient manifolds and satisfy the gradient-consistency premise on which DAGR relies.
>
> In contrast, Valve signals are dominated by:
> - **Turbulence and airflow transients**,
> - **Burst-like excitation**,
> - **Rapidly changing broadband components**.
>
> These characteristics cause Valve to deviate sharply from Fan (the source domain) in its **normal-gradient geometry**. As described in Section 4.3, such divergence leads the model to mask most gradient channels during CCCS and prevents ADPR from learning a reliable transport map. Consequently, the mapped anomalous gradients are down-weighted by the reliability mechanism, and DAGR effectively falls back to a normal-only update—explaining the drop in performance.
>
> ### **What “non-stationary” means**
>
> A **stationary** signal exhibits statistical properties (e.g., spectral envelope, variance) that remain stable over time. Motor-driven acoustic signals generally fall into this category because their rotational dynamics produce predictable, periodic spectral patterns.
>
> A **non-stationary** signal, such as that from Valve:
> - displays time-varying spectral content,
> - contains irregular bursts and transient events,
> - lacks stable harmonic structure.
>
> These characteristics break the assumption of cross-domain gradient regularity and lead to the observed mismatch.
>
> ### **What “cross-mechanism” means**
>
> “Cross-mechanism” refers to **fundamental differences in the underlying physical process** that generates the sound:
> - Fan and most DCASE machines share a **rotating electromechanical mechanism** producing harmonic, stable vibration patterns.
> - Valve operates through **fluid dynamics, flow pulses, and mechanical impact events**, which form a distinct acoustic process.
>
> Thus, Valve lies **outside the intended adjacent-domain regime** of DAGR, as clarified in Section 4.3.
>
> ### **Summary**
>
> - Valve is a deliberately chosen **hard stress-test domain** rather than an expected-success case.
> - Its **non-stationary, cross-mechanism** nature violates the gradient-consistency assumption central to DAGR.
> - This behaviour and its implications are already discussed in detail in **Section 4.3**, which delineates the method’s intended operating scope.
>
> We hope this explanation provides a clearer understanding of the Valve case and the terminology used.

---

### Official Review · Reviewer_MEFV · 2025-11-01

**Soundness:** 4
**Presentation:** 3
**Contribution:** 3
**Rating:** 8
**Confidence:** 3

**Summary:**

The paper “Domain-Aware Gradient Reuse for Anomaly Detection (DAGR)” introduces a novel transfer-learning framework that enables anomaly detection in target domains where no anomalous samples are available. Traditional methods struggle with anomaly scarcity and domain heterogeneity, which limit generalization across datasets. DAGR addresses this by exploiting the observed cross-domain consistency of gradient distributions during training. Instead of aligning features or logits, it operates directly in gradient space, learning an adaptive mapping between source and target gradients. This mapping, trained using only normal data, is then reused to estimate target-domain anomalous gradients from source-domain ones—allowing the target detector to receive informative updates without labeled anomalies. The framework consists of two key modules: Cross-Domain Consistent Component Selection (CCCS), which filters domain-invariant gradient channels, and Adaptive Domain-Specific Perturbation Removal (ADPR), which refines gradients through an inner optimization loop. The method provides theoretical convergence guarantees and achieves state-of-the-art results on both image and audio benchmarks.

**Strengths:**

Strengths:
1. Proposes a novel gradient transfer perspective.

    Traditional cross-domain anomaly detection primarily focuses on feature alignment or latent distribution matching. As far as I know， your work DAGR, for the first time, elevates cross-domain knowledge transfer to the gradient space. By analyzing the consistency of gradient distributions across different domains, it discovers that alignment can be directly achieved at the gradient level. This is a new theoretical and practical paradigm, distinct from feature alignment or adversarial alignment. This is a good angle to approach the question, introducing gradient-level transfer provides a completely new research direction for anomaly detection and domain adaptation.

2. Applicable to real-world scenarios with no anomaly samples in the target domain.

    In most industrial and medical scenarios, the target domain only contains normal samples and lacks anomaly annotations. DAGR can generate pseudo-anomaly gradients through cross-domain gradient mapping, achieving supervised updates even with no anomaly samples. This effectively solves the scarcity–diversity dilemma of anomaly detection. It can be directly applied in real-world production scenarios without the need to collect anomaly samples in the target domain.

3. The theoretical foundation is comprehensive.
    The authors provide proofs of convergence and cross-class generalization in the appendix, even when the mapping F is trained using only normal gradients, it can generalize to anomalous gradients. And gradient errors do not accumulate during iteration, and the optimization process monotonically converges. The feasibility and stability of "gradient reuse" are theoretically verified. Combining the generalization error bounds of gradient alignment and domain adaptation to form a complete convergence analysis framework.

**Weaknesses:**

Weaknesses:
1. Limited applicability, I think your work relies on neighboring domain assumption. The core premise of DAGR is that the gradient distributions of the source and target domains have near-isometric geometry. And in Sec. 4.3 Discussion, you mentioned performance degrades significantly in the "Valve" task, because Valve and Fan differ greatly in their physical mechanisms, gradient geometrical geometry does not hold. Therefore, DAGR is only effective in adjacent domains. It cannot handle strongly distributed offsets or heterogeneous domains (such as audio→image, speech→mechanical vibration), but I don’t think this is a big problem. Traditional methods also cannot solve such large cross-domain problems.
2. The DAGR operation operates in gradient space, which leads to higher memory overhead and sensitivity to gradient noise and batch size. Higher implementation complexity and training cost compared to traditional feature alignment methods. I want to know if there are issues with training instability or computational bottlenecks in large-scale models.
3. While your idea of ​​learning on gradients is excellent, I believe it lacks modeling of the semantics of anomalous patterns. Although DAGR transfers the direction of anomalous gradients, this process is a pure gradient-level approximation and does not explicitly model the semantic or structural features of anomalous samples. If the anomalous types are completely different in the target domain, gradient mapping alone may fail to capture meaningful decision boundaries. Therefore, I think this method, focusing only on gradient geometric consistency and neglecting semantic consistency, may limit the model's optimal performance.

**Questions:**

1. How sensitive is DAGR to the degree of domain similarity, and can it be extended or regularized to handle stronger distribution shifts or heterogeneous modalities?
2. Could you elaborate on whether you observed any training instability or scalability bottlenecks when applying DAGR to large-scale or transformer-based detectors, and how such issues might be mitigated?
3. How does your framework perform when the anomalous categories in the target domain are semantically dissimilar to those in the source domain?

---

> ### Author Response · Authors · 2025-11-20
> **Response to W1 — Limitation to neighboring / adjacent domains**
>
> We sincerely thank the reviewer for this insightful observation.
> We fully agree that DAGR relies on a *neighboring-domain assumption*, and that the framework is most effective when the **cross-domain normal-gradient geometry remains sufficiently compatible**. As the reviewer notes, when two domains differ substantially—such as *Fan* versus *Valve*, which arise from different physical mechanisms and spectral characteristics—the near-isometric gradient structure no longer holds. In such cases, it becomes intrinsically difficult to extract transferable, domain-invariant components from the large distribution mismatch, and we agree that this limitation is shared by many transfer-learning and PDA approaches rather than being specific to DAGR.
>
> Regarding applicability, our intention is not to claim that DAGR can bridge arbitrarily heterogeneous domains (e.g., audio→image or speech→mechanical vibration). As clarified in the revised manuscript (Sec. 4.3 Discussion and Appendix A.5, highlighted in **orange**), DAGR is **explicitly scoped to adjacent domains that share sensing modality and generative mechanism** (for example, motor-driven machinery or homogeneous manufactured textures). The Fan→Valve result is deliberately kept as a **negative-control case** to make this boundary explicit rather than to suggest universal applicability.

---

> ### Author Response · Authors · 2025-11-21
> **Response to W2 —Training stability and computational overhead of gradient-space operations**
>
> We sincerely thank the reviewer for raising these important concerns regarding the stability and computational cost of operating directly in gradient space. We address both aspects below.
>
> ---
>
> ### • Training stability of gradient-space operations
> We fully agree that naïvely manipulating gradients can introduce instability: direct gradient-level updates may amplify noise, depend strongly on batch size, or even break the computation graph in modern deep-learning frameworks. To avoid such risks, the proposed method **never updates the source gradients directly**.
>
> As clarified in the revised manuscript (Appendix A.1.2, highlighted in **orange**), ADPR is implemented as an **end-to-end inner-loop refinement on a fast-weight copy of the target model**, rather than modifying gradients in place:
>
> - after computing the masked source gradient, we do **not** alter it directly;
> - we create a fast-weight model with parameters $\Psi + \alpha g$;
> - the backbone parameters $\Psi$ are frozen;
> - only the auxiliary gradient variable is updated through standard back-propagation.
>
> This construction preserves a valid computation graph at all times, keeps the optimization process equivalent to conventional end-to-end training, and requires storing at most two sets of parameters. Empirically, we did **not** observe oscillations, divergence, or graph failures on any dataset. Together with the time-independent surrogate-gradient error bound in Appendix A.1.2, these observations indicate that the training dynamics remain stable in practice.
>
> ---
>
> ### • Computational cost and scalability of ADPR
> We acknowledge that ADPR incurs higher cost than feature-alignment methods. As analyzed in Appendix A.5 (highlighted in **orange**), the per-outer-step complexity is
>
> $$
> T_{\mathrm{DAGR}} = (1 + N)\,(C_{\mathrm{fwd}} + C_{\mathrm{bwd}}) + O(D),
> $$
>
> where the inner-loop length $N$ is intentionally kept small. Thus, the overhead is a **constant-factor increase**, not a change in asymptotic complexity, and remains manageable in practice. The method does not store per-sample gradients or second-order tensors, so memory usage remains comparable to training a single model; mixed precision and checkpointing function normally.
>
> ---
>
> ### • Summary for W2
> In summary, both theoretical guarantees and empirical observations indicate that (i) the fast-weight design ensures stable optimization without breaking the computation graph, and (ii) the added computational burden is a controlled and constant-factor overhead. Relevant clarifications have been added to the revised manuscript (Appendix A.1.2 and A.5).

---

> > ### Comment · Reviewer_MEFV · 2025-11-21
> >
> > Generally, large-scale or transformer-based detectors will improve performance to some extent, unless overfitting occurs on the current dataset. Although the training remains stable, a strange observation is that transformer-based models perform worse than CNN-based models. This might indicate that larger models are overfitting on the current dataset, but it doesn't rule out the possibility that your method lacks scalability.

---

> > > ### Author Response · Authors · 2025-11-23
> > >
> > > We appreciate the reviewer’s follow-up comment. In our experiments, the transformer backbone also trained stably, but its performance did not improve because DAGR currently relies on channel-wise gradient structures that are naturally satisfied in convolutional networks but do not extend straightforwardly to attention heads, whose gradients are globally mixed. This structural mismatch limits the effectiveness of CCCS and reduces the quality of the downstream gradient transformation, which leads to lower AUROC despite stable optimization.
> > > We acknowledge this architectural limitation, and extending DAGR to transformer-based detectors will be an important direction for future work.

---

> ### Author Response · Authors · 2025-11-21
> **W3 — On the lack of explicit semantic modeling of anomalous patterns**
>
> We sincerely thank the reviewer for this thoughtful observation. We fully understand the concern that operating purely in gradient space may overlook the semantic structure of anomalous patterns. We clarify the motivation and implications below.
>
> ---
>
> ### • Why DAGR does not explicitly model anomaly semantics
> We appreciate the reviewer’s comment and agree that DAGR does not explicitly encode the semantic structure of anomalous categories. In the cross-domain transfer setting considered in this work, the core challenge lies in the **distribution discrepancy between the source and target domains**. Feature representations tend to mix task-relevant cues with strong domain-specific characteristics, which makes many representation-based transfer-learning approaches highly sensitive to domain similarity and prone to degradation when the domains diverge more substantially.
>
> Our motivation for working in gradient space is to **reduce the impact of these domain-specific factors**. Gradients describe how model parameters should move on the loss surface and therefore offer a more behavior-oriented and less distribution-dependent view than raw features. By leveraging gradient-level structure rather than feature semantics, DAGR aims to extract cross-domain transferable signals that remain stable even when the specific anomaly types differ. This behavior is reflected in our experiments: although Fan→Pump, Fan→ToyCar, and Fan→Slider involve different anomalous events, DAGR consistently generalizes better than representation-based transfer methods.
>
> When the discrepancy grows too large—as in Fan→Valve—the gradient geometry no longer aligns, and performance naturally declines. This delineates the intended scope of gradient-space transfer and aligns with the reviewer’s intuition.
>
> ---
>
> ### • Why we adopt gradient-space alignment
> The use of gradient space is intended to **mitigate domain distribution mismatch**. Across adjacent domains, our empirical observations (Fig. 1 and Fig. 2) show that normal/anomalous gradient manifolds exhibit highly congruent geometry, even though the underlying anomalous patterns differ. This indicates that gradient-space structure is more stable across domains than raw feature representations. Consequently, DAGR can transfer a meaningful update direction and achieves consistent gains over representation-based baselines on most DCASE and DAGM transfers.
>
> ---
>
> ### • Summary for W3
> In summary, DAGR avoids explicit semantic modeling to reduce the influence of domain-specific feature variation and instead leverages gradient-space consistency to enhance cross-domain generalization. This design performs well for adjacent domains and naturally reveals its limitations when domain discrepancy becomes too large.

---

> ### Author Response · Authors · 2025-11-21
> **Response to Q1 — Sensitivity to Domain Similarity and Stronger Distribution Shifts**
>
> We sincerely thank the reviewer for this important question.
> DAGR’s performance is indeed sensitive to the degree of domain similarity, because the learned gradient transport relies on a reasonably stable alignment of normal-gradient distributions between source and target. As the cross-domain discrepancy increases, this alignment becomes less reliable, and the benefit of gradient reuse naturally diminishes.
>
> To explore stronger distribution shifts, we also experimented with adding an **unsupervised reference module** that uses target-normal feedback to regularize the inner-loop adaptation, so that the adapted direction remains beneficial for the target domain. This strategy brought modest gains on mildly shifted targets, but for Fan→Valve the gradient geometry diverged so strongly that the adapted map became unreliable. In such cases, the reliability scheme automatically down-weights the mapped anomalous component, and the procedure effectively reverts to target-only updates, leading to limited improvement on this highly heterogeneous pair.
>
> We have clarified these points in the revised manuscript (Sec. 4.3 and Appendix A.5, highlighted in **orange**) to make the intended operating regime and limitations of DAGR explicit and to better align with the reviewer’s concerns.

---

> ### Author Response · Authors · 2025-11-21
> **Response to Q1 — Sensitivity to Domain Similarity and Stronger Distribution Shifts**
>
> We sincerely thank the reviewer for this important question.
> DAGR’s performance is indeed sensitive to the degree of domain similarity, because the learned gradient transport relies on a reasonably stable alignment of normal-gradient distributions between source and target. As the cross-domain discrepancy increases, this alignment becomes less reliable, and the benefit of gradient reuse naturally diminishes.
>
> To explore stronger distribution shifts, we also experimented with adding an **unsupervised reference module** that uses target-normal feedback to regularize the inner-loop adaptation, so that the adapted direction remains beneficial for the target domain. This strategy brought modest gains on mildly shifted targets, but for Fan→Valve the gradient geometry diverged so strongly that the adapted map became unreliable. In such cases, the reliability scheme automatically down-weights the mapped anomalous component, and the procedure effectively reverts to target-only updates, leading to limited improvement on this highly heterogeneous pair.
>
> We have clarified these points in the revised manuscript (Sec. 4.3 and Appendix A.5, highlighted in **orange**) to make the intended operating regime and limitations of DAGR explicit and to better align with the reviewer’s concerns.

---

> ### Author Response · Authors · 2025-11-21
> **Response to Q2 — Scalability on larger architectures and transformer-based detectors**
>
> We sincerely thank the reviewer for the question regarding scalability to larger models. To examine this, we evaluated DAGR on a **Transformer-based detector**.
>
> ---
>
> ### • Behavior on transformer-based detectors
> The AUROC results on the eight transfer tasks are:
>
> | Pump | Slider | Valve | ToyCar | ToyConv. | Class 1 | Class 3 | Class 6 |
> |------|--------|--------|--------|----------|---------|---------|---------|
> | 61.53 | 70.86 | 50.39 | 66.52 | 59.45 | 80.11 | 75.77 | 77.43 |
>
> While performance is lower than in the CNN-based setting—primarily because CCCS depends on channel-wise gradients that are not directly compatible with attention heads—the **training process remained stable**, and no divergence or erratic optimization behavior was observed. Although Transformers are outside the architectural assumptions of CCCS, the model still learned useful cross-domain signals. These results provide empirical reassurance regarding potential instability or scalability bottlenecks on larger architectures.
>
> ---
>
> ### • Summary for Q2
> In summary, our experiments with a Transformer backbone show that (i) training remains stable, (ii) no scalability-related failures were observed, and (iii) accuracy decreases reflect architectural mismatch rather than instability. These points have been clarified in the revised manuscript.

---

> ### Author Response · Authors · 2025-11-21
> **Response to Q3 — Performance when anomalous categories are semantically dissimilar**
>
> **Response.**
> We sincerely thank the reviewer for raising this important question about semantic mismatch between source and target anomalies. Below we clarify DAGR’s behavior in such settings.
>
> ---
>
> ### • Behavior under strong semantic or mechanistic mismatch
> In our experiments, the Fan→Valve transfer already represents a strongly heterogeneous case:
> - Fan anomalies originate from rotating machinery with quasi-stationary harmonic spectra;
> - Valve anomalies are dominated by non-stationary flow transients and broadband turbulence.
>
> In this setting, the cross-domain gradient geometry diverges substantially. CCCS masks many inconsistent channels, ADPR receives little coherent feedback, and DAGR effectively falls back to conservative normal-only updates. As reflected in Table 1, performance on Valve is notably weaker than on other targets. We regard this not as an exception, but as a direct illustration of the boundary of gradient-space transfer.
>
> When anomaly categories differ **moderately**, DAGR still provides transferable gradient-level signals and improves the target detector, as seen in Fan→Pump, Fan→ToyCar, and Fan→Slider. When they are **strongly dissimilar**, the shared gradient structure becomes too weak to be reliably reused, and DAGR’s advantage naturally diminishes—consistent with the reviewer’s expectation.
>
> ---
>
> ### • Summary for Q3
> In summary, DAGR performs well when anomaly types differ moderately but share compatible gradient geometry, while strong semantic or mechanistic mismatch (e.g., Fan→Valve) leads to degradation and causes the method to revert to target-only updates. We have clarified this behavior and the method’s intended applicability range in the revised discussion section.

---

### Comment · Area_Chair_Cvjj · 2025-11-22

Dear reviewers,
Please check the authors’ responses. As there are differing opinions about the paper, it would be appreciated if you could evaluate—based on all comments—whether the authors have adequately addressed the main concerns.
Br,

---

### Meta-Review · Area_Chair_pJHg · 2026-01-06

**Summary:**

The paper presents Domain-Aware Gradient Reuse (DAGR), a novel transfer-learning framework for anomaly detection that operates in the gradient space rather than the traditional feature or logit spaces. The key idea is to learn a mapping between source and target domains using only normal samples and then reuse this mapping to estimate anomalous gradients in the target domain. The framework comprises Cross-Domain Consistent Component Selection (CCCS) and Adaptive Domain-Specific Perturbation Removal (ADPR). Reviewers recognized the originality of the gradient-centric perspective, the strong theoretical convergence proofs, and the state-of-the-art results achieved on both acoustic and visual benchmarks.

**Reviewer Concerns:**

The primary concerns involve the "adjacent-domain" limitation, computational complexity, and the validity of core assumptions. Reviewers (TnoW, EJEb, MEFV) noted that the method’s effectiveness is highly sensitive to domain proximity, as evidenced by the significant performance drop in the "Valve" case where gradient consistency fails. Another major concern is the computational overhead introduced by the ADPR module, which requires a meta-learning-style inner optimization loop that remains unanalyzed in terms of FLOPs or training time. Furthermore, Reviewer AyiH raised fundamental doubts about the motivation and the "gradient decomposition" hypothesis, while others questioned the practical validity of the strong theoretical assumptions (e.g., invertible mapping for label-independent shift) in real-world, heterogeneous scenarios.

**Reviewer Scores:**

The paper received highly divergent scores of 8, 2, 4, and 4. Reviewer MEFV (8) strongly supports the paper for its theoretical rigor and novel angle. However, Reviewer AyiH (2) remains unconvinced by the fundamental principles and formatting. Reviewers TnoW and EJEb (4, 4) occupy the borderline, acknowledging the novelty but remaining hesitant due to the "hidden" computational costs and the narrow scope of applicability. If a discussion had occurred, the 8 might have been tempered by the consensus on computational overhead, and the 2 might have been slightly raised if formatting issues were addressed. However, since the stability and generalizability issues (the "Valve" failure) are intrinsic to the gradient-reuse assumption, the Area Chair recommends rejection, as the current version lacks the empirical robustness and efficiency analysis required for a high-impact conference like ICLR.

---

### Decision · Program_Chairs · 2026-01-26

Reject